# DIFFUSION SAMPLING WITH MOMENTUM FOR MITIGATING DIVERGENCE ARTIFACTS

**Suttisak Wizadwongsa**
VISTEC, Thailand
suttisak.w_s19@vistec.ac.th

**Worameth Chinchuthakun**
Tokyo Institute of Technology, Japan
chinchuthakun.w.aa@m.titech.ac.jp

**Pramook Khungurn**
pixiv Inc.
pramook@gmail.com

**Amit Raj**
Google
amitrajs@google.com

**Supasorn Suwajanakorn**
VISTEC, Thailand
supasorn.s@vistec.ac.th

## ABSTRACT

Despite the remarkable success of diffusion models in image generation, slow sampling remains a persistent issue. To accelerate the sampling process, prior studies have reformulated diffusion sampling as an ODE/SDE and introduced higher-order numerical methods. However, these methods often produce *divergence* artifacts, especially with a low number of sampling steps, which limits the achievable acceleration. In this paper, we investigate the potential causes of these artifacts and suggest that the small stability regions of these methods could be the principal cause. To address this issue, we propose two novel techniques. The first technique involves the incorporation of Heavy Ball (HB) momentum, a well-known technique for improving optimization, into existing diffusion numerical methods to expand their stability regions. We also prove that the resulting methods have first-order convergence. The second technique, called Generalized Heavy Ball (GHVB), constructs a new high-order method that offers a variable trade-off between accuracy and artifact suppression. Experimental results show that our techniques are highly effective in reducing artifacts and improving image quality, surpassing state-of-the-art diffusion solvers on both pixel-based and latent-based diffusion models for low-step sampling. Our research provides novel insights into the design of numerical methods for future diffusion work.

## 1 INTRODUCTION

Diffusion models (Ho et al., 2020; Song et al., 2020a) are a family of generative models that has garnered considerable attention due to their remarkable image quality. Unlike Generative Adversarial Networks (GANs) (Goodfellow et al., 2014), which may suffer from mode collapse and instabilities during training, diffusion models are less sensitive to hyperparameters (Ho et al., 2020; Rombach et al., 2022) and offer better sampling quality (Dhariwal & Nichol, 2021). These models have been successfully applied to various image-related tasks, such as text-to-image generation (Nichol et al., 2022), image-to-image translation (Su et al., 2022), image composition (Sasaki et al., 2021), adversarial purification (Wang et al., 2022; Wu et al., 2022), and super-resolution (Choi et al., 2021).

However, one significant drawback of diffusion models is their slow sampling speed. This is because the sampling process involves a Markov chain, which requires a large number of iterations to generate high-quality results. Recent attempts to accelerate this process include improvements to the noise schedule (Nichol & Dhariwal, 2021; Watson et al., 2021) and network distillation (Salimans & Ho, 2022; Watson et al., 2022; Song et al., 2023). Fortunately, the sampling process can be represented by ordinary or stochastic differential equations, and numerical methods can be used to reduce the number of iterations required. While DDIM (Song et al., 2020a), a $1^{st}$-order method, is the most commonly used approach, it still requires a considerable number of iterations. Higher-order numerical methods, such as DEIS (Zhang & Chen, 2022), DPM-Solver (Lu et al., 2022a), and PLMS (Liu et al., 2022a), have been proposed to generate high-quality images in fewer steps. However, these

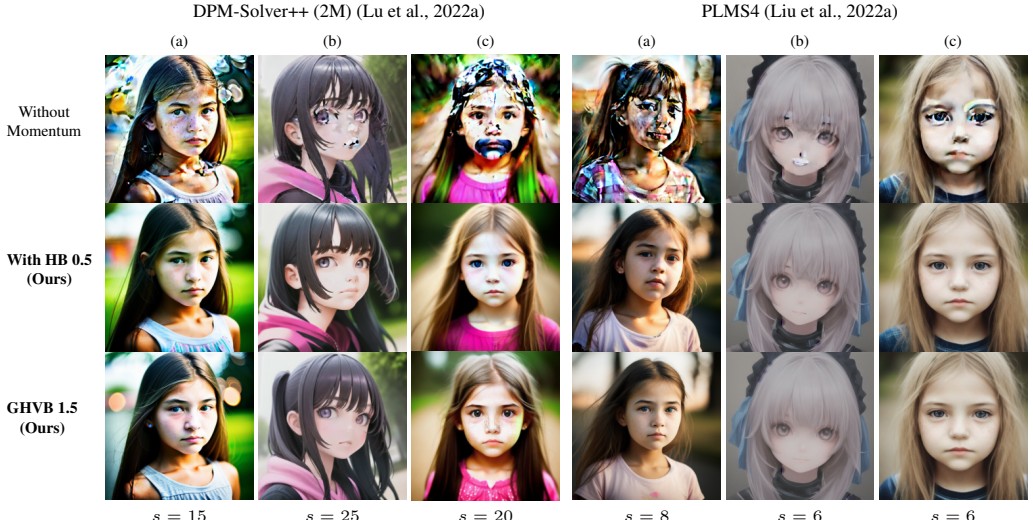

Figure 1: We demonstrate the occurrence of divergence artifacts in existing numerical methods at various guidance scales $s$ using 15 sampling steps. Integrating HB momentum into these methods and our GHVB 1.5 method can mitigate these artifacts. Prompt: "photo of a girl face" (a) Realistic Vision v2.0, (b) Anything Diffusion v4.0, (c) Deliberate Diffusion.

methods begin to produce artifacts (see Figure 1) when the number of steps is decreased beyond a certain value, thereby limiting how much we can reduce the sampling time.

In this study, we investigate the potential causes of these artifacts and found that the narrow stability region of high-order numerical methods can cause solutions to diverge, resulting in divergence artifacts. To address this issue and enable low-step, artifact-free sampling, we propose two techniques. The first technique involves incorporating Heavy Ball (HB) momentum (Polyak, 1987), a well-known technique for improving optimization, into existing diffusion numerical methods. This approach effectively reduces divergence artifacts, but its accuracy only has first order of convergence. In this context, the accuracy measures how close the approximated, low-step solution is to the solution computed from a very high-step solver (e.g., 1,000-step DDIM). The second technique, called Generalized Heavy Ball (GHVB), is a new high-order numerical method that offers a variable trade-off between accuracy and artifact suppression. Both techniques are training-free and incur negligible additional computational costs. Figure 1 demonstrates the capability of both techniques in reducing artifacts compared to previous diffusion sampling methods. Furthermore, our experiments show that our techniques are effective on both pixel-based and latent-based diffusion models.

## 2 BACKGROUND

This section first presents the theoretical foundation of diffusion sampling when modeled as an ordinary differential equation (ODE) and related numerical methods. Second, we discuss ODE forms for guided diffusion sampling and prior splitting numerical methods. Third, we review the concept of stability region, which is our primary analysis tool.

### 2.1 DIFFUSION IN ODE FORM

Modeling diffusion sampling as an ODE is commonly based on the non-Markovian sampling of Denoising Diffusion Implicit Model (DDIM) from Song et al. (2020a). DDIM is given by:

$$x_{t-1} = \sqrt{\frac{\alpha_{t-1}}{\alpha_t}} \left( x_t - \sqrt{1 - \alpha_t} \epsilon_\theta(x_t, t) \right) + \sqrt{1 - \alpha_{t-1}} \epsilon_\theta(x_t, t). \tag{1}$$

Here, $\epsilon_\theta(x_t, t)$ is a neural network with learnable parameters $\theta$ trained to predict noise given the current state $x_t$ and time $t$ as input. The parameter $\alpha_t$ is a schedule that controls the degree of diffusion at each time step. Previous research has shown that DDIM (1) can be rewritten into an

ODE, making it possible to use numerical methods to accelerate the sampling process. Two ODEs have been proposed in the literature:

$$\frac{d\bar{x}}{d\sigma} = \bar{\epsilon}(\bar{x}, \sigma), \qquad (2) \qquad\qquad \frac{d\tilde{x}}{d\tilde{\sigma}} = s(\tilde{x}, \tilde{\sigma}), \qquad (3)$$

Equation 2 can be obtained by re-parameterizing $\sigma = \sqrt{1 - \alpha_t}/\sqrt{\alpha_t}$, $\bar{x} = x_t/\sqrt{\alpha_t}$, and $\bar{\epsilon}(\bar{x}, \sigma) = \epsilon_\theta(x_t, t)$. These transformations are widely used in various diffusion solvers (Zhang & Chen, 2022; Lu et al., 2022a; Liu et al., 2022a; Zhao et al., 2023). If $\epsilon_\theta(x_t, t)$ is a sum of multiple terms, such as in guided diffusion, we can split Equation 2 and solve each resulting equation separately (Wizadwongsa & Suwajanakorn, 2023). Another ODE (Equation 3) is derived by defining $\tilde{\sigma} = \sqrt{\alpha_t}/\sqrt{1 - \alpha_t}$ and $\tilde{x} = x_t/\sqrt{1 - \alpha_t}$, where an estimation of the final result $s(\tilde{x}, \tilde{\sigma}) = (x_t - \sqrt{1 - \alpha_t}\epsilon_\theta(x_t, t))/\sqrt{\alpha_t}$. This ODE has the advantage of keeping the differentiation bounded within the pixel value range in pixel-based diffusion, outperforming Equation 2 in many cases in DPM-Solver++ (Lu et al., 2022b).

## 2.2 STABILITY REGION

The stability region is a fundamental concept in numerical methods for solving ODEs. It determines the step sizes that enable numerical approximations to converge. To illustrate this concept, let us consider the Euler method, a simple, $1^{\text{st}}$-order method for solving ODEs, given by

$$x_{n+1} = x_n + \delta f(x_n), \qquad (4)$$

where $x_n$ is the approximate solution and $\delta$ is the step size. To analyze the stability of the Euler method, we can consider a test equation of the form $x' = \lambda x$, where $\lambda$ is a complex constant. The solution of this test equation can be expressed as

$$x_{n+1} = x_n + \delta\lambda x_n = (1 + \delta\lambda)x_n = (1 + \delta\lambda)^{n+1}x_0, \qquad (5)$$

where $x_0$ is the initial value. For the approximate solution to converge to the true solution, it is necessary that $|1 + \delta\lambda| \leq 1$. Hence, the stability region of the Euler method is $S = \{z \in \mathbb{C} : |1 + z| \leq 1\}$ because if $z = \delta\lambda$ lies outside of $S$, the solution $x_n$ will tend to $\pm\infty$ as $n \to \infty$.

In diffusion sampling, another common family of numerical solvers is the Adams-Bashforth (AB) methods, also referred to as Pseudo Linear Multi-Step (PLMS). AB methods encompass the Euler method as its $1^{\text{st}}$-order (AB1), and the $2^{\text{nd}}$-order (AB2) is given by:

$$x_{n+1} = x_n + \delta\left(\frac{3}{2}f(x_n) - \frac{1}{2}f(x_{n-1})\right). \qquad (6)$$

The stability regions of AB methods of various orders are derived in Appendix C. These regions can be visualized by determining their boundaries using the boundary locus technique (Lambert et al., 1991). As depicted in Figure 2, the stability region decreases in size and its boundary becomes more restrictive as the order of the method increases.

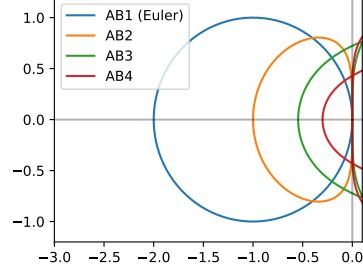

Figure 2: Boundaries of stability regions of the first 4 Adams-Bashforth methods.

## 3 UNDERSTANDING ARTIFACTS IN DIFFUSION SAMPLING

One unique issue in diffusion sampling is the occurrence of "divergence" artifacts, which are characterized by regions with unrealistic, oversaturated pixels in the outputs. This problem can arise from several factors including the use of high-order numerical solvers, too few sampling steps, or a high guidance scale. (See Appendix P). The existing solution is simply avoiding these factors at the cost of slower sampling speed or weaker guidance. This section investigates the cause of these artifacts in Section 3.1 and proposes solutions that do not sacrifice sampling speed in Section 3.2.

## 3.1 ANALYZING DIFFUSION ARTIFACTS

We analyze the areas where divergence artifacts occur during sampling by examining the magnitudes of the latent variables in those areas. Specifically, we use Stable Diffusion (Rombach et al., 2022),

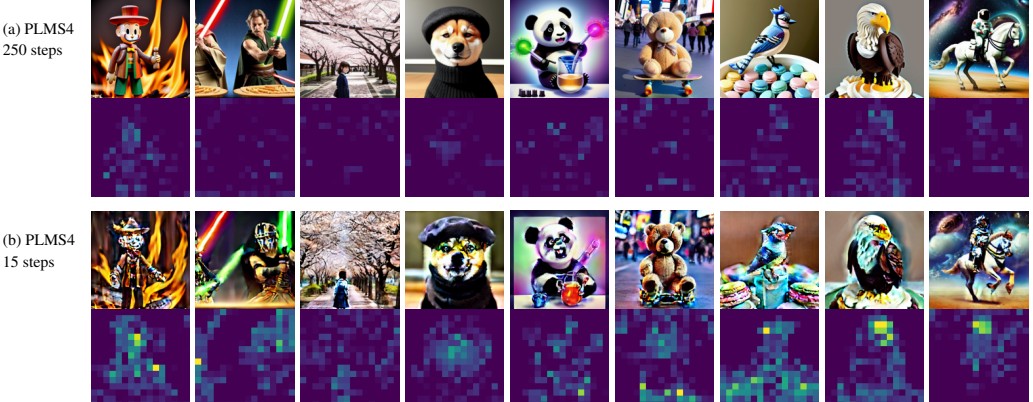

Figure 3: Generated images and max-pooled latent magnitude maps (brighter colors indicate higher values). Images with artifacts (b) have higher latent magnitudes than those without artifacts (a).

which operates and performs diffusion sampling on a latent space of dimension $64 \times 64 \times 4$, to generate images with and without artifacts by varying the number of steps. Then, we visualize each latent variable $z \in \mathbb{R}^4$ in the $64 \times 64$ spatial grid after normalizing it using the channel-wise mean and standard deviation, computed from the COCO dataset (Lin et al., 2014). Figure 3 shows the magnitudes of the normalized latent variables after a $4 \times 4$ max pooling for visualization purposes.

We found that artifacts mainly appear in areas where the latent magnitudes are higher than usual. Note that images without artifacts can also have high latent magnitudes in some regions, although this is exceptionally rare. Conversely, when artifacts appear, those regions almost always have high magnitudes. In pixel-based diffusion models, the artifacts manifest directly as pixel values near 1 or 0 due to clipping, which can be observed in Figure 17 in Appendix K.

## 3.2 CONNECTION BETWEEN ODE SOLVER AND ARTIFACTS

We hypothesize that these artifacts are caused by the numerical instability during sampling. Mathematically, we analyze the ODE for diffusion sampling in Equation 2 using the problem reduction technique for stiffness analysis (Higham & Trefethen, 1993). Assuming that the effect of $\sigma$ on the function $\bar{\epsilon}$ is negligible, we use Taylor expansion to approximate the RHS. of Equation 2, yielding

$$\frac{d\bar{x}}{d\sigma} = \nabla\bar{\epsilon}(x^*)(\bar{x} - x^*) + \mathcal{O}(\|\bar{x} - x^*\|^2).  \tag{7}$$

Here, $x^*$ denotes the converged solution that should not have any noise left (i.e. $\bar{\epsilon}(x^*) = 0$), and $\nabla\bar{\epsilon}(x^*)$ denotes the Jacobian matrix at $x^*$. As $\bar{x}$ converges to $x^*$, the term $\mathcal{O}(\|\bar{x} - x^*\|^2)$ becomes negligibly small, so we may drop it from the equation.

Let $\lambda$ be an eigenvalue of $\nabla\bar{\epsilon}(x^*)^T$ and $v$ be the corresponding normalized eigenvector such that $\nabla\bar{\epsilon}(x^*)^T v = \lambda v$. We define $u = v^T(\bar{x} - x^*)$ and obtain $u' = \lambda u$ as our test equation. According to Section 2.2, if $\delta\lambda$ falls outside the stability region of a numerical method, the numerical solution to $u$ may diverge, resulting in diffusion sampling results with larger magnitudes that later manifest as divergence artifacts. Therefore, divergence artifacts are more likely to occur when the stability region is too small. Although some numerical methods have infinite stability regions, those used in diffusion sampling have only finite stability regions, which implies that the solution will always diverge if the step size $\delta$ is sufficiently high. More details about the derivation can be found in Appendix D and a 2D toy example illustrating this effect is provided in Appendix E.

One possible solution to mitigate artifacts is to reduce the step size $\delta$, which shifts $\delta\lambda$ closer to the origin of the complex plane. However, this approach increases the number of steps, making the process slower. Instead, we will modify the numerical methods to enlarge their stability regions.

## 4 METHODOLOGY

This section describes two techniques for improving stability region and reducing divergence artifacts. Specifically, we first show how to apply Polyak's Heavy Ball Momentum (HB) to diffusion sampling, and secondly, how to generalize it to higher orders. Our techniques are designed to be simple to implement and do not require additional training.

### 4.1 POLYAK'S HEAVY BALL MOMENTUM FOR DIFFUSION SAMPLING

Recall that Heavy Ball Momentum is an optimization algorithm proposed by Polyak (1987) that enhances gradient descent ($x_{n+1} = x_n - \beta_n \nabla f(x_n)$). The method takes inspiration from the physical analogy of a heavy ball moving through a field of potential with damping friction. The update rule for Polyak's HB optimization algorithm is given by:

$$x_{n+1} = x_n + \alpha_n(x_n - x_{n-1}) - \beta_n \nabla f(x_n), \tag{8}$$

where $\alpha_n$ and $\beta_n$ are parameters. We can apply HB to the Euler method (4), in which case we typically set $\alpha_n = (1 - \beta_n)$, to obtain

$$x_{n+1} = x_n + (1 - \beta_n)(x_n - x_{n-1}) + \delta\beta_n f(x_n), \tag{9}$$

and we may show that the numerical method above has the same order of convergence as the original Euler method. For simplicity, we assume that $\beta_n = \beta \in (0, 1]$, which is a constant known as the *damping coefficient.* Then, we can reformulate Equation 9 as:

$$v_{n+1} = (1 - \beta)v_n + \beta f(x_n), \qquad x_{n+1} = x_n + \delta v_{n+1}, \tag{10}$$

Here, we may interpret $x_n$ as the heavy ball's position, and $v_{n+1}$—the exponential moving average of $f(x_n)$—as its velocity. We can see that position is updated with "displacement = time × velocity," much like in physics.

For a high-order method of the form $x_{n+1} = x_n + \delta \sum_{i=0}^{k} b_i f(x_{n-i})$, we can apply HB to it as:

$$v_{n+1} = (1 - \beta)v_n + \beta \sum_{i=0}^{k} b_i f(x_{n-i}), \qquad x_{n+1} = x_n + \delta v_{n+1}. \tag{11}$$

The resulting numerical method has a larger stability region, as can be seen in Figures 4b to 4d, in which we show stability boundaries of AB methods after HB is applied to them with varying $\beta$s. (We use HB 0.4 to denote $\beta = 0.4$). However, Theorem 1 in Appendix H shows that as soon as $\beta$ deviates from 1, the theoretical order of convergence drops to 1, leading to a significant decrease in image quality, as illustrated in Figure 5. In the next subsection, we propose an alternative approach that increases the stability region while maintaining high order of convergence.

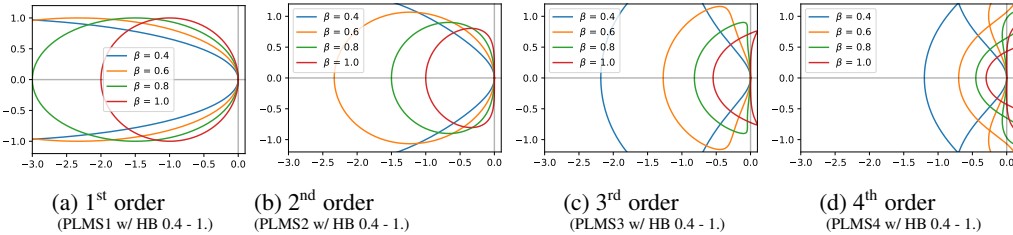

(a) 1$^{st}$ order
(PLMS1 w/ HB 0.4 - 1.)

(b) 2$^{nd}$ order
(PLMS2 w/ HB 0.4 - 1.)

(c) 3$^{rd}$ order
(PLMS3 w/ HB 0.4 - 1.)

(d) 4$^{th}$ order
(PLMS4 w/ HB 0.4 - 1.)

Figure 4: Boundaries of stability regions of 1$^{st}$- to 4$^{th}$-order AB methods with HB applied to them with different values of the damping coefficient $\beta$.

### 4.2 GENERALIZING POLYAK'S HEAVY BALL TO HIGHER ORDERS

In this section, we generalize the Euler method with HB momentum to achieve high-order convergence, similar to AB methods' generalization of the Euler method. Defining the backward difference operator $\Delta$ as $\Delta x_n = x_n - x_{n-1}$. Following Berry & Healy (2004), we express the AB formula by:

$$\Delta x_{n+1} = \delta \left(1 + \frac{1}{2}\Delta + \frac{5}{12}\Delta^2 + \frac{3}{8}\Delta^3 + \frac{251}{720}\Delta^4 + \frac{95}{288}\Delta^5 + \dots\right) f(x_n). \tag{12}$$

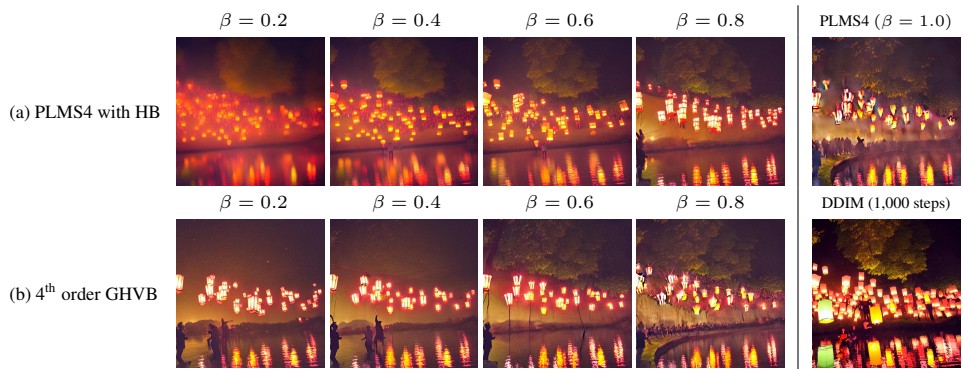

Figure 5: Comparison between the two proposed techniques: (a) HB and (b) GHVB, applied to PLMS4 (Liu et al., 2022a) with 15 sampling steps. Both are effective at reducing artifacts, but HB's accuracy drops faster than GHVB's as $\beta$ deviates from 1. Positions of the lanterns in Row (a) deviate more from the ground-truth (1,000-steps DDIM (Zhang et al., 2023)) than those in Row (b). Moreover, the image at $\beta = 0.2$ in Row (a) becomes blurry as HB yields a numerical method with a lower order of convergence than what GHVB does. Prompt: "A beautiful illustration of people releasing lanterns near a river".

The order convergence is determined by the number of terms on the RHS; for example, AB2 can be written as $\Delta x_{n+1} = \delta \left(1 + \frac{1}{2}\Delta\right) f(x_n)$. Multiplying Equation 12 by $(\beta + (1 - \beta)\Delta)$, we have:

$$(\beta + (1 - \beta)\Delta)\Delta x_{n+1} = \delta \left(\beta + \frac{2 - \beta}{2}\Delta + \frac{6 - \beta}{12}\Delta^2 + \frac{10 - \beta}{24}\Delta^3 + \dots\right) f(x_n). \quad (13)$$

Note that the update of $v_n$ in Equation 10 can be rewritten as $(\beta + (1 - \beta)\Delta)v_{n+1} = \beta f(x_n)$. Next, we can choose the order of convergence by fixing the number of terms on the RHS. To get, say, a 2nd-order method, we may choose:

$$(\beta + (1 - \beta)\Delta)\Delta x_{n+1} = \delta \left(\beta + \frac{2 - \beta}{2}\Delta\right) f(x_n) = \delta \left(1 + \frac{2 - \beta}{2\beta}\Delta\right) \beta f(x_n) \quad (14)$$

$$= \delta \left(1 + \frac{2 - \beta}{2\beta}\Delta\right)(\beta + (1 - \beta)\Delta)v_{n+1}. \quad (15)$$

Eliminating $(\beta - (1 - \beta)\Delta)$ from both sides, we obtain the 2nd-order generalized HB method:

$$v_{n+1} = (1 - \beta)v_n + \beta f(x_n), \qquad x_{n+1} = x_n + \delta \left(\frac{2 + \beta}{2\beta}v_{n+1} + \frac{2 - \beta}{2\beta}v_n\right). \quad (16)$$

Algorithm 2 details a complete implementation. When $\beta = 1$, the formulation in Equation 16 is equivalent to the AB2 formulation in Equation 6. As $\beta$ approaches 0, Equation 14 converges to the 1st-order Euler method 4. Thus, this generalization also serves as an interpolating technique between two adjacent-order AB methods, except for the 1st-order GHVB, which is equivalent to the Euler method with HB momentum in Equation 10.

We call this new method the Generalized Heavy Ball (GHVB) and associate with it a *momentum number*, whose ceiling indicates the method's order. For example, GHVB 1.8 refers to the 2nd-order GHVB with $\beta = 0.8$. The main difference between HB and GHVB is that HB calculates the moving average after summing high-order coefficients, whereas GHVB calculates it before the summation.

We analyze the stability region of GHVB using the same approach as before and visualize the region's locus curve in Figure 6. The theoretical order of accuracy of this method is given by Theorem 2 in Appendix H. We discuss alternative momentum methods, such as Nesterov's momentum, which can offer comparable performance but are less simple in Appendix G.

## 5 EXPERIMENTS

We present a series of experiments to evaluate the effectiveness of our techniques. In Section 5.1, we assess the divergence artifacts reduction through qualitative results and quantitative measurements

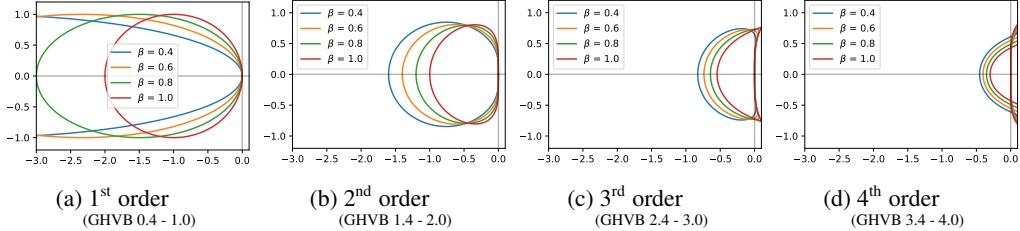

|  |  |  |  |
|---|---|---|---|
| (a) 1$^{st}$ order | (b) 2$^{nd}$ order | (c) 3$^{rd}$ order | (d) 4$^{th}$ order |
| (GHVB 0.4 - 1.0) | (GHVB 1.4 - 2.0) | (GHVB 2.4 - 3.0) | (GHVB 3.4 - 4.0) |

Figure 6: Boundary of stability regions for 1$^{st}$-to 4$^{th}$- order Generalized Heavy Ball (GHVB).

of the latent magnitudes in a text-to-image diffusion model. Besides reducing artifacts, another important goal is to ensure that the overall sampling quality improves and does not degenerate (e.g., becoming color blobs). We test this with experiments on both pixel-based and latent-based diffusion models trained on ImageNet256 (Russakovsky et al., 2015) (Section 5.2 and 5.3), which show that our techniques significantly improve image quality, as measured by the standard Fréchet Inception Distance (FID) score. Lastly, in Section 5.4, we present an ablation study of GHVB methods with varying degrees of order. A similar study on HB methods can be found in Appendix Q.

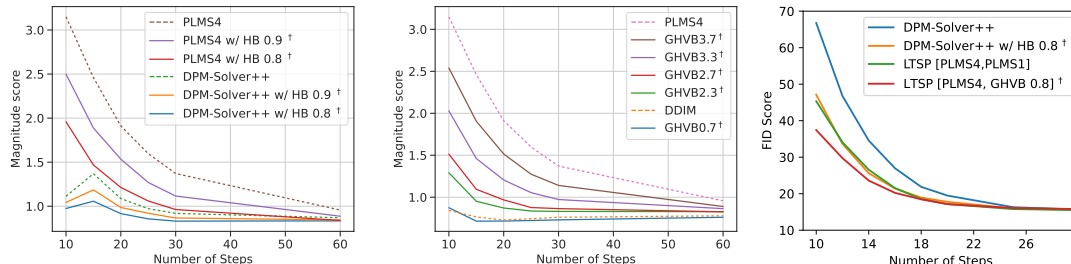

Figure 7: Mean magnitude scores   Figure 8: Mean magnitude scores   Figure 9: FID on ADM (†ours)

## 5.1 ARTIFACTS MITIGATION

In this experiment, we apply our HB and GHVB techniques to the most popular 2$^{nd}$ and 4$^{th}$-order solvers, DPM-Solver++ (Lu et al., 2022a) and PLMS4 (Liu et al., 2022a), using 15 sampling steps and various guidance scales on three different text-to-image diffusion models. The qualitative results in Figure 1 show that our techniques significantly reduce the divergence artifacts and produce realistic results (columns a, c). More qualitative results are in Figure 16 in Appendix J.

Quantitatively measuring divergence artifacts can be challenging, as metrics like MSE or LPIPS may only capture the discrepancy between the approximated and the true solutions, which does not necessarily indicate the presence of divergence artifacts. In this study, we use the magnitudes of latent variables as introduced in Section 3.1 as a proxy metric to measure artifacts. In particular, we define a magnitude score $v = \sum_{i,j} g(z'_{ij})$ that sums over the latent variables in a max-pooled latent grid, where $g(x) = x$ if $x \geq \tau$ and 0 otherwise. We generate 160 samples from the same set of text prompts and seeds for each method from a fine-tuned Stable Diffusion model called Anything V4.

The results using $\tau = 3$ (magnitude considered high when above 3 std.) are shown in Figures 7 and 8. We observe that the magnitude score increases as the number of sampling steps decreases and higher-order methods result in higher magnitude scores. Adding HB momentum to PLMS4 (Liu et al., 2022a) or DPM-Solver++ (Lu et al., 2022b) (Figure 7) or employing GHVB with momentum number $\beta < 1$ (Figure 8) can reduce these latent magnitudes. Our methods also show a larger performance difference from previous works as the problem setup is more challenging, as demonstrated in additional Stable Diffusion experiments in Appendix M and O.

Our recommendation is to employ the Euler method with HB 0.8 or GHVB 0.8 as the default choices for common usage due to their consistent performance and sufficiently expansive stability regions (see Appendix for more experiments). In scenarios where accuracy or fidelity to the true DDIM solution is essential or when a lower sampling step is required, higher-order GHVB variants can be considered. For guidance on method selection, please refer to Q2 in Appendix A.

## 5.2 EXPERIMENTS ON PIXEL-BASED DIFFUSION MODELS

We evaluate our techniques using classifier-guided diffusion sampling with ADM from Peebles & Xie (2022), an unconditioned pixel-based diffusion model, with their classifier model. Additionally, we compare our methods with two other diffusion sampling methods, namely DPM-Solver++ (Lu et al., 2022b) and LTSP (Wizadwongsa & Suwajanakorn, 2023), which have demonstrated strong performance in classifier-guided sampling.

For DPM-Solver++, we use a 2nd-order multi-step method and compare the results with and without HB momentum. For LTSP, a split numerical method, we use PLMS4 (Liu et al., 2022a) to solve the first subproblem (see (Wizadwongsa & Suwajanakorn, 2023)) and compare different methods for solving the second subproblem, including regular Euler method and its variant with HB momentum (equivalent to GHVB 0.8). Figure 9 indicates that our techniques effectively improve FID scores for both DPM-Solver++ and LTSP. Notably, applying our HB momentum to LTSP consistently yields the lowest FID scores. This experiment highlights the benefits of using HB momentum, which provides a better choice than Euler method. Table 1 presents additional results, and Figure 17 provides examples of the generated images.

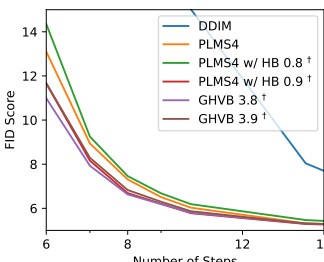

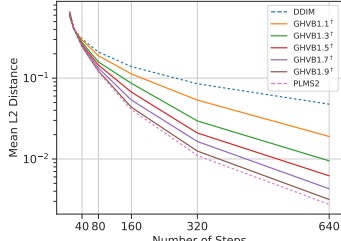

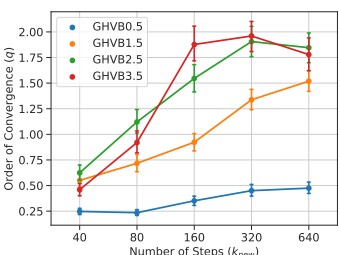

Figure 10: FID score on DiT. (†ours)

Figure 11: L2 between different samplers to 1,000-step DDIM.

Figure 12: Numerical order of convergence for GHVB.

## 5.3 EXPERIMENT ON LATENT-BASED DIFFUSION MODELS

We evaluate our techniques using classifier-free guidance diffusion sampling with pre-trained latent-space diffusion models: DiT-XL (Peebles & Xie, 2022) and MDT (Gao et al., 2023) (more in Appendix N). We use the 4th-order PLMS4 as our baseline for comparison because it demonstrates superior performance compared to other methods (see Appendix L).

In Figure 10, a significant gap in FID scores on DiT can be observed between 4th-order PLMS4 and 1st-order DDIM, but this is mostly due to the difference in convergence order rather than divergence artifacts. Our GHVB 3.8 and 3.9 techniques successfully mitigate numerical divergence and lead to improved FID scores compared to PLMS4, particularly when the number of steps is below 10. While HB 0.9 also improves FID scores, using HB 0.8 can worsen FID scores compared to using PLMS4 alone, as it shares 1st-order convergence with DDIM. For high sampling steps, both HB and GHVB achieve comparable performance to PLMS4 without significant quality degradation. We provide additional results in Appendix L, and example images in Figure 18.

## 5.4 ABLATION STUDY OF GHVB

In this section, we conduct an ablation study of the GHVB method. As explained in Section 4.2, the damping coefficient $\beta$ of GHVB interpolates between two existing AB methods, DDIM and PLMS2. Our goal here is to analyze the convergence error of GHVB methods. The comparison is done on Stable Diffusion 1.5 with the target results obtained from a 1,000-step DDIM method. We measure the mean L2 distance between the sampled results and the target results in the latent space. The results in Figure 11 suggest that the convergence error of GHVB 1.1 to GHVB 1.9 interpolates between the convergence errors of DDIM and PLMS2 accordingly.

Furthermore, we empirically verify that GHVB does achieve a high order of convergence as predicted by Theorem 2. We compute the numerical order of convergence using the formula $q \approx \frac{\log(e_{\text{new}}/e_{\text{old}})}{\log(k_{\text{new}}/k_{\text{old}})}$, where $e$ is the error between the sampled and the target latent codes, and $k$ is the

number of sampling steps. As shown in Figure 12, the numerical orders of GHVB 0.5 and GHVB 1.5 approach 0.5 and 1.5, respectively, as the number of steps increases. However, for GHVB 2.5 and GHVB 3.5, the estimated error $e$ may be too small when tested with large numbers of steps, and other sources of error may hinder their convergence. Nonetheless, these GHVB methods can achieve high orders of convergence. An analysis of this and other methods is in Appendix I.

## 6    RELATED WORK

**Accelerating diffusion sampling**: Our work is closely related to several other approaches aimed at improving the sampling speed of diffusion models. Certain approaches focus on the improved numerical aspects of samplers, which we have already covered in Section 2.1. Other approaches involve training additional separate models, including model distillation (Luhman & Luhman, 2021; Salimans & Ho, 2022), Schrödinger bridge (De Bortoli et al., 2021), consistency models (Song et al., 2023), rectified flow (Liu et al., 2022b), and GENIE (Dockhorn et al., 2022b). Our goal is an ODE solver that seamlessly integrates with any pre-trained diffusion models. This focus sets our work apart from those aforementioned approaches, which require model retraining.

**Connections between momentum methods and ODEs**: Our work uses an optimization technique to create an ODE solver. Interestingly, there is a rich body of research in the near-opposite direction: using ODEs to analyze optimization methods. Indeed, trajectories made by some optimization methods can be described as discrete approximations of ODE solutions. The gradient descent algorithm is equivalent to numerically integrating the gradient flow equation: $x' = -\nabla f(x(t))$. Polyak (1964) introduced the Heavy Ball method and the Heavy Ball ODE, $x'' + ax' + b\nabla f(x(t)) = 0$, which is subsequently investigated by Alvarez (2000); Attouch et al. (2000); Balti & May (2016); Shi et al. (2021), and Goudou & Munier (2009). Similarly, Nesterov (1983) introduced the accelerated gradient algorithm and derived its associated ODE, given by: $x'' + \frac{a}{t}x' + \nabla f(x(t)) = 0$. Further studies (Su et al., 2014; Bubeck et al., 2015; Allen-Zhu & Orecchia, 2014; Lessard et al., 2016) analyze similar ODEs to understand Nesterov's algorithm better.

Notice that the above momentum associated ODEs are $2^{\text{nd}}$-order. Nonetheless, we can also regard momentum methods as a numerical approximation of the $1^{\text{st}}$-order gradient flow, which is very similar to our equation. (Our ODE (2) can be obtained simply by replacing the gradient with the diffusion model's output.) Analyzing our solvers through the $2^{\text{nd}}$-order ODEs is less fruitful as there is no guarantee that their trajectories would match those of our DDIM's ODE—a key goal in our work, inspired by DDIM's excellent quality (De Bortoli et al., 2021; Karras et al., 2022).

**Momentum in neural network design**: The benefits of Heavy Ball ODE, which result in smoother paths and faster convergence to stationary points, have inspired research on neural network architecture design (Moreau & Bruna, 2017; Nguyen et al., 2020; Li et al., 2018) and $2^{\text{nd}}$-order neural ODE models (Xia et al., 2021; Nguyen et al., 2022). In the context of diffusion models, Wu et al. (2023) reformulate a diffusion ODE into a $2^{\text{nd}}$-order ODE to enhance the training and sampling processes, whereas Dockhorn et al. (2022a) does so to the score-based SDE. Instead of redesigning the network with momentum, our paper specifically proposes new momentum-based solvers for the normal diffusion network to improve its sampling process.

## 7    CONCLUSION

Our study highlights an issue when using high-order methods in diffusion sampling with low sampling steps, which leads to the divergence of solutions and artifacts. To address this, we propose two techniques inspired by Polyak's HB momentum: HB and GHVB. HB serves as an add-on to any diffusion solver to reduce artifacts, while GHVB offers flexibility in selecting the most suitable method for any computation budget. Our extensive experiments confirm the effectiveness of both techniques across image generation tasks with recent diffusion models, such as text-to-image generation (Stable Diffusion (Rombach et al., 2022)), classifier guidance (ADM (Dhariwal & Nichol, 2021)) and classifier-free sampling (on DiT (Peebles & Xie, 2022), and on MDT (Gao et al., 2023)).

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

# Part I

# Appendices

## Contents

## A   FREQUENTLY ASKED QUESTIONS

### Q1: What does the term "divergence artifacts" refer to?

In this paper, the term "divergence artifacts" is used to describe visual anomalies that occur when numerical solutions *diverge*, resulting in unusually large magnitudes of the results. In the context of latent-based diffusion, we specifically define divergence artifacts as visual artifacts caused by latent codes with magnitudes that exceed the usual range. These artifacts commonly arise when the stability region of the numerical method fails to handle all eigenvalues of the system, leading to a divergent numerical solution. We demonstrate the connection between high latent magnitudes and these artifacts in Figure 13.

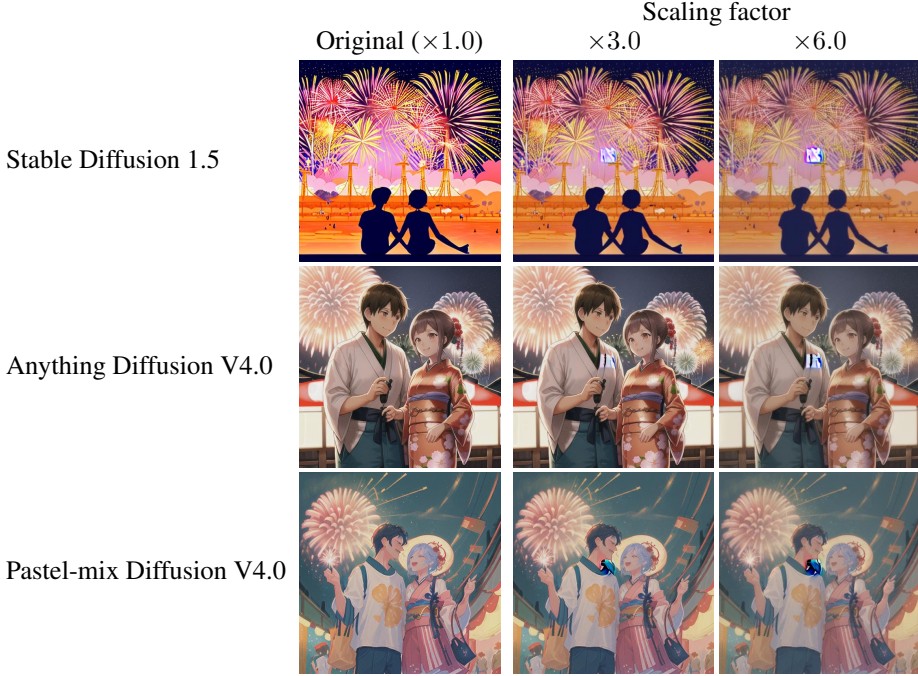

Figure 13: Images generated after multiplying the $4 \times 4$ square at the center of latent codes with a constant factor. Using higher scaling factors results in more apparent localized divergence artifacts. All samples were generated using PLMS4 (Liu et al., 2022a) with a guidance scale of 15 and 250 sampling steps. Prompt: "A beautiful illustration of a couple looking at fireworks in a summer festival in Japan".

### Q2: How to efficiently determine a suitable solver for diffusion sampling?

Theoretically identifying a solver with optimal accuracy and a stability region that encompasses all $\delta\lambda$ values is challenging because computing $\lambda$ from neural networks is intractable. However, given a fixed amount of computational resources (e.g., sampling steps), one can gather information about

$\lambda$ to determine the highest solver's order without artifacts by first testing Euler's method with fewer numbers of steps.

In particular, we can determine the lowest sampling step of Euler's method that does not produce artifacts (e.g., via a binary search), then use its stability region to determine the highest order of GHVB whose stability region encompasses that of Euler's method. This process does not require additional sampling or evaluating neural networks and can be solved deterministically (Equation 48-51). This strategy is more efficient than directly testing different orders at the target step because Euler's method usually requires fewer steps than our target step. For instance, we can infer that GHVB3.2 and those with lower orders are applicable at 20 sampling steps by verifying that Euler's method works at 5 steps.

### Q3: Can we directly interpolate two existing numerical methods instead of using the GHVB method?

Indeed, this is possible. However, the order of the resulting method will be the lowest order of the two methods. To illustrate this point, let us consider a direct interpolation between the $1^{\text{st}}$-order Euler method (AB1) and the $2^{\text{nd}}$-order Adams-Bashford method (AB2), expressed as follows:

$$x_{n+1} = x_n + \delta \left( (1-\beta)f(x_n) + \beta\frac{3}{2}f(x_n) - \beta\frac{1}{2}f(x_{n-1}) \right) \tag{17}$$

As outlined in Appendix H, despite the orders of the interpolated methods, the resulting method is a $1^{\text{st}}$-order method.

## B   MORE DETAILS ON DIFFUSION MODELING AND SAMPLING

This section presents a high-level summary of the theoretical foundation of diffusion models, along with the numerical methods employed in these models. Here, we briefly explain a few concepts that contribute to our method.

### B.1   DIFFUSION MODELS AND THEIR SAMPLING

Assuming that $x_0$ is a random variable from the data distribution $q(x_0)$ that we aim to replicate (e.g., the distribution of natural images), diffusion models define a sequence of Gaussian noise degradations of $x_0$ as random variables $x_1, x_2, ..., x_T$, where

$$x_t \sim q(x_t|x_{t-1}) = \mathcal{N}(\sqrt{1-\beta_t}x_{t-1}, \beta_t\mathbf{I}) \tag{18}$$

and $\beta_t \in [0, 1]$ are parameters controlling the noise levels. Utilizing the property of the Gaussian distribution, we express $x_t$ directly as a function of $x_0$ and noise $\epsilon \sim \mathcal{N}(0, \mathbf{I})$ by

$$x_t = \sqrt{\bar{\alpha}_t}x_0 + \sqrt{1-\bar{\alpha}_t}\epsilon, \tag{19}$$

where $\bar{\alpha}_t = \prod_{i=1}^{t}(1-\beta_i)$. By selecting a sufficiently large $T$ (e.g., $T = 1,000$) and an appropriate set of $\beta_t$, we can assume that $x_T$ follows a standard Gaussian distribution $\mathcal{N}(0, \mathbf{I})$.

The main idea of diffusion model generation involves training $p_\theta(x_{t-1}|x_t)$, which sequentially reverses $q(x_t|x_{t-1})$ using a parametric model while setting $p(x_t) = \mathcal{N}(0, \mathbf{I})$. Subsequently, we can sample Gaussian noise $x_T$ and use it to reverse-sample $x_{T-1}, x_{T-2}, ...$ until we retrieve $x_0$, which belongs to our data distribution.

The Denoising Diffusion Probabilistic Model (DDPM) Ho et al. (2020) outlines the utilization of a neural network denoted as $\epsilon_\theta(x_t, t)$ to predict the noise $\epsilon$ used in the computation of $x_t$ in Equation 19. To facilitate the network's training, a training image labeled as $x_0$ is sampled, alongside values for $t$ and $\epsilon$. These provided values are then utilized in computing $x_t$ using the aforementioned relationship. Consequently, our network $\epsilon_\theta$ is optimized to minimize the difference between the projected noise and the actual noise: $\|\epsilon - \epsilon_\theta(x_t, t)\|^2$. Once training is completed, we can draw samples $x_{t-1}$ from the conditional distribution $p_\theta(x_t|x_{t-1})$ given by

$$p_\theta(x_t|x_{t-1}) = \mathcal{N}\left( \mu_\theta(x_{t-1}, t), \frac{1-\bar{\alpha}_{t-1}}{1-\bar{\alpha}_t}\mathbf{I} \right), \tag{20}$$

where $\mu_\theta(x_{t-1}, t) = \frac{1}{\sqrt{\bar{\alpha}_t}}\left(x_t - \frac{\beta_t}{\sqrt{1-\bar{\alpha}_t}}\epsilon_\theta(x_t, t)\right)$ and $x_T \sim p(x_T) = \mathcal{N}(0, \mathbf{I})$. This step-by-step sampling process is simple but time-consuming. Several sampling techniques and ODE formulations can be employed to accelerate this process. (See Section 2.1.)

Many successful diffusion models Dhariwal & Nichol (2021); Song & Ermon (2020); Song et al. (2020b) consider their target distributions $q_0(x_0)$ as image distributions, with $x_t$ directly representing a random variable within the pixel space. These models generate images by inferring networks directly through diffusion sampling. However, direct training of diffusion models within high-resolution pixel space are computationally expensive. To address this, Latent Diffusion Models (LDMs) Rombach et al. (2022) employ a two-stage methodology: Firstly, they employ a trained pair of encoder $E$ and decoder $D$ that compress images into smaller spatial latent representations. Secondly, LDMs are then trained on these latent representations $z = E(x)$, rather than directly on images $x$. Subsequently, the generation of new images involves sampling a latent representation $z$ from the diffusion model and decoding it into an image using the decoder, $x = D(z)$.

## B.2 NUMERICAL METHOD SUMMARY

Many numerical methods can solve Equations 2 and 3 to accelerate diffusion sampling. In the following section, we will summarize the methods mentioned within our papers. Let's denote our ODE as $\frac{dx}{d\sigma} = f(x)$, initialized at $x_0$.

**Euler's Method** is represented by

$$x_{n+1} = x_n + \delta f(x_n), \tag{21}$$

where $x_n$ represents the approximate solution, and $\delta$ represents the step size. This method, when applied to Equation 2 or 3, results in the DDIM formulation Song et al. (2020a).

**Heun's Method** is a 2nd-order extension of Euler's Method given by:

$$x_{n+1} = x_n + \frac{\delta}{2}(e_1 + e_2), \tag{22}$$

Here, $e_1 = f(x_n)$ and $e_2 = f(x_n + \delta e_1)$. This method has been used in various diffusion papers, including Algorithm 1 in Karras et al. (2022) and DPM-Solver-2 in Lu et al. (2022a), and also serves as the simplest form of Predictor-Corrector methods in Song et al. (2020b); Zhao et al. (2023).

**Adams-Bashforth (AB) Methods** is a family of methods utilizing previous steps to estimate the next step, also known as linear multi-step methods. The formulations are as follows:

**1st-order** (same as Euler's method):

$$x_{n+1} = x_n + \delta f(x_n), \tag{23}$$

**2nd-order**:

$$x_{n+1} = x_n + \frac{\delta}{2}\left(3e_0 - e_1\right), \tag{24}$$

**3rd-order**:

$$x_{n+1} = x_n + \frac{\delta}{12}(23e_0 - 16e_1 + 5e_2), \tag{25}$$

**4th-order**:

$$x_{n+1} = x_n + \frac{\delta}{24}(55e_0 - 59e_1 + 37e_2 - 9e_3), \tag{26}$$

where $e_k = f(x_{n-k})$. Notably, DPM-Solver++ Lu et al. (2022b) applies the 2nd-order AB method to ODE (3), while PLMS4 Liu et al. (2022a) utilizes the 4th-order AB method on ODE (2).

## B.3 GUIDED DIFFUSION SAMPLING

Guided diffusion sampling is widely used in conditional sampling, such as text-to-image and class-to-image generation. There are main two approaches for guided sampling:

**Classifier guidance** (Dhariwal & Nichol, 2021; Song et al., 2020a) uses a pre-trained classifier model $p_\phi(c \mid x_t, t)$ to define the conditional noise prediction model at inference time:

$$\hat{\epsilon}(x_t, t \mid c) = \epsilon_\theta(x_t, t) - s\nabla \log p_\theta(c \mid x_t, t), \tag{27}$$

where $s > 0$ is a "guidance" scale. The model can be extended to accept any guidance function, such as CLIP function (Radford et al., 2021) for text-to-image generation (Letts et al., 2021). This approach only modifies the sampling equation at inference time and thus can be applied to a trained diffusion model without retraining.

**Classifier-free guidance** (Ho & Salimans, 2021) trains a conditional noise model $\epsilon_\theta(x_t, t \mid c)$ to generate data samples with the label $c$:

$$\hat{\epsilon}(x_t, t \mid c) = \epsilon_\theta(x_t, t \mid \phi) + s(\epsilon_\theta(x_t, t \mid c) - \epsilon_\theta(x_t, t \mid \phi)), \tag{28}$$

where $\phi$ is a null label to allow for unconditional sampling. The sampling equations in both approaches can be expressed as a "guided ODE" of the form

$$\frac{d\bar{x}}{d\sigma} = \bar{\epsilon}(\bar{x}, \sigma) + g(\bar{x}, \sigma), \tag{29}$$

where $g(\bar{x}, \sigma)$ represents a guidance function. To accelerate guided diffusion sampling, splitting numerical methods have been proposed by Wizadwongsa & Suwajanakorn (2023), such as Lie-Trotter Splitting (LTSP) divides Equation 29 into two subproblems, i) $\frac{dy}{d\sigma} = \bar{\epsilon}(y, \sigma)$ and ii) $\frac{dz}{d\sigma} = g(z, \sigma)$, but only apply high-order numerical methods to the first equation while resorting to the Euler method for the second equation to avoid numerical instability. Higher-order splitting methods, such as Strang Splitting (STSP), can also mitigate artifacts. However, these methods require solving the second equation twice per step, which is comparable to increasing the total sampling step to avoid artifacts. Both approaches require non-negligible computation.

## C   STABILITY REGION OF ADAM-BASHFORTH METHOD

To investigate the stability of the AB2 method, we apply AB2 to the test equation $x' = \lambda x$, which was also used with the Euler method (Section 2.2). We have $x_{n+1} = x_n + \delta\left(\frac{3}{2}\lambda x_n - \frac{1}{2}\lambda x_{n-1}\right)$. To solve this linear recurrence relation, we substitute $x_n = r^n$ into the formula, where $r$ is a complex constant. Simplifying the resulting equation, we obtain the characteristic equation:

$$r^2 - \left(1 + \frac{3}{2}\delta\lambda\right)r + \frac{1}{2}\delta\lambda = 0, \tag{30}$$

which has the solutions

$$r_1 = \frac{1}{2}\left(1 + \frac{3}{2}\delta\lambda + \sqrt{\left(1 + \frac{3}{2}\delta\lambda\right)^2 - 2\delta\lambda}\right), \tag{31}$$

$$r_2 = \frac{1}{2}\left(1 + \frac{3}{2}\delta\lambda - \sqrt{\left(1 + \frac{3}{2}\delta\lambda\right)^2 - 2\delta\lambda}\right). \tag{32}$$

The general formulation of $x_n$ can be expressed as

$$x_n = a_1 r_1^n + a_2 r_2^n, \tag{33}$$

where $a_1$ and $a_2$ are constants. The numerical solution $x_n$ tends to 0 as $n$ tends to infinity when both $|r_1| < 1$ and $|r_2| < 1$, which means the stability region of AB2 is determined by the complex region

$$S = \left\{z \in \mathbb{C} : \left|\frac{1}{2}\left(1 + \frac{3}{2}z \pm \sqrt{\left(1 + \frac{3}{2}z\right)^2 - 2z}\right)\right| \leq 1\right\}. \tag{34}$$

Solving for the complex area from the roots of the characteristic equation can pose significant challenges in numerical analysis. One commonly employed graphical technique to visualize the stability region is the boundary locus technique (Lambert et al., 1991).

## C.1 THE BOUNDARY LOCUS TECHNIQUE

The boundary locus technique begins by defining the shift operator $E$ such that $Ex_k = x_{k-1}$. Note that $E^2 x_k = Ex_{k-1} = x_{k-2}$. Generally, a numerical method can be represented in the following form:

$$A(E)x_n = \delta B(E)f(x_n), \tag{35}$$

where $A$ and $B$ are polynomials of $E$. For example, in the case of the AB2 method, we have $A(E) = 1 - E$ and $B(E) = \frac{3}{2}E - \frac{1}{2}E^2$.

To determine the stability region of a numerical method, we apply the boundary locus technique to the general form given by Equation 35. The characteristic equation of the method can be obtained by substituting $f(x_n) = \lambda x_n$ (i.e., the test equation) and $x_n = r^n$, which yields

$$A(r^{-1}) = \delta \lambda B(r^{-1}), \tag{36}$$

where $r$ is the root of the method's characteristic equation. The stability region of the method is the area in the complex plane where the characteristic root $r$ have modulus less than 1. The boundary of the stability region can be determined by substituting $r$ with a modulus of 1 (which means that $r = e^{i\theta}$ for some real value $\theta$) into the characteristic equation and solving for $z = \delta\lambda$. This yields the locus of points in the complex plane where the characteristic roots of the method are on the boundary of the stability region. Specifically, we can obtain the curve $z = s(\theta) = A(e^{-i\theta})/B(e^{-i\theta})$, where $\theta \in [-\pi, \pi]$, that represents the boundary of the stability region in the complex plane. By comparing the stability regions of different numerical methods, we can determine which method is more stable and accurate for a given problem. The boundary locus technique provides a powerful tool for analyzing the stability of numerical methods and can help guide the selection of appropriate methods for solving ODE problems.

**Example C.1.** *(Euler Method) The Euler method, a numerical technique for approximating solutions of ODE, can be expressed as:*

$$(1 - E)x_n = \delta E f(x_n) \tag{37}$$

*The associated polynomials for this method are:*

$$A(z) = 1 - z, \quad B(z) = z \tag{38}$$

*The stability region of the Euler method corresponds to the locus curve in which the solution remains bounded. This region can be determined by evaluating the complex function:*

$$s(\theta) = \frac{A(e^{-i\theta})}{B(e^{-i\theta})} = \frac{1 - e^{-i\theta}}{e^{-i\theta}} = e^{i\theta} - 1, \quad \theta \in [-\pi, \pi]. \tag{39}$$

*The locus curve forms a perfect circle with a radius of 1 and a center at -1.*

**Example C.2.** *(AB Methods) The $2^{nd}$-order Adams-Bashforth (AB2) method is given by:*

$$(1 - E)x_n = \delta \left( \frac{3}{2}E - \frac{1}{2}E^2 \right) f(x_n). \tag{40}$$

*The locus curve representing the stability region of this method is given by:*

$$s(\theta) = \frac{1 - e^{-i\theta}}{\frac{3}{2}e^{-i\theta} - \frac{1}{2}e^{-2i\theta}} = \frac{2(1 - e^{-i\theta})}{3e^{-i\theta} - e^{-2i\theta}}, \quad \theta \in [-\pi, \pi]. \tag{41}$$

*Similarly, the stability regions for the AB3 and AB4 methods can be obtained by evaluating the complex functions:*

$$s(\theta) = \frac{12(1 - e^{-i\theta})}{23e^{-i\theta} - 16e^{-2i\theta} + 5e^{-3i\theta}}, \quad \theta \in [-\pi, \pi]. \tag{42}$$

$$s(\theta) = \frac{24(1 - e^{-i\theta})}{55e^{-i\theta} - 59e^{-2i\theta} + 37e^{-3i\theta} - 9e^{-4i\theta}}, \quad \theta \in [-\pi, \pi]. \tag{43}$$

*The locus curves for the boundary of stability regions of the first four AB methods are visualized in Figure 2.*

Additionally, we include the locus curves for the boundaries of the stability regions of our techniques. As shown in Figure 4, the local curves for the PLMS method with HB $\beta$ are given by:

**PLMS1 with HB $\beta$:**

$$s(\theta) = \frac{(1 - e^{-i\theta})(1 - (1 - \beta)e^{-i\theta})}{\beta e^{-i\theta}} \tag{44}$$

**PLMS2 with HB $\beta$:**

$$s(\theta) = \frac{2(1 - e^{-i\theta})(1 - (1 - \beta)e^{-i\theta})}{\beta(3e^{-i\theta} - e^{-2i\theta})} \tag{45}$$

**PLMS3 with HB $\beta$:**

$$s(\theta) = \frac{12(1 - e^{-i\theta})(1 - (1 - \beta)e^{-i\theta})}{\beta(23e^{-i\theta} - 16e^{-2i\theta} + 5e^{-3i\theta})} \tag{46}$$

**PLMS4 with HB $\beta$:**

$$s(\theta) = \frac{24(1 - e^{-i\theta})(1 - (1 - \beta)e^{-i\theta})}{\beta(55e^{-i\theta} - 59e^{-2i\theta} + 37e^{-3i\theta} - 9e^{-4i\theta})} \tag{47}$$

Similarly, the local curves for the GHVB method in Figure 6 are given by:

**1$^{\text{st}}$-order GHVB** (equivalent to PLMS1 with HB):

$$s(\theta) = \frac{(1 - e^{-i\theta})(1 - (1 - \beta)e^{-i\theta})}{\beta e^{-i\theta}} \tag{48}$$

**2$^{\text{nd}}$-order GHVB**:

$$s(\theta) = \frac{2(1 - e^{-i\theta})(1 - (1 - \beta)e^{-i\theta})}{((2 + \beta)e^{-i\theta} - (2 - \beta)e^{-2i\theta})} \tag{49}$$

**3$^{\text{rd}}$-order GHVB**:

$$s(\theta) = \frac{12(1 - e^{-i\theta})(1 - (1 - \beta)e^{-i\theta})}{(18 + 5\beta)e^{-i\theta} - (24 - 8\beta)e^{-2i\theta} + (6 - \beta)e^{-3i\theta}} \tag{50}$$

**4$^{\text{th}}$-order GHVB**:

$$s(\theta) = \frac{24(1 - e^{-i\theta})(1 - (1 - \beta)e^{-i\theta})}{(46 + 9\beta)e^{-i\theta} - (78 - 19\beta)e^{-2i\theta} + (42 - 5\beta)e^{-3i\theta} - (10 - \beta)e^{-4i\theta}} \tag{51}$$

## D  DERIVATION OF TEST EQUATION

This section presents the derivation of the test equation $u' = \lambda u$, which serves as a fundamental tool for analyzing the stability of numerical methods in diffusion sampling, as discussed in Section 3.2.

Starting from the differential equation for $\bar{x}$, we have

$$\frac{d\bar{x}}{d\sigma} = \nabla\bar{\epsilon}(x^*)(\bar{x} - x^*). \tag{52}$$

We then define $u = v^T(\bar{x} - x^*)$, where $v$ is a normalized eigenvector of $\nabla\bar{\epsilon}(x^*)^T$ corresponding to the eigenvalue $\lambda$. Taking the derivative of $u$ with respect to $\sigma$ and using the chain rule, we have:

$$\frac{du}{d\sigma} = v^T \frac{d}{d\sigma}(\bar{x} - x^*) = v^T[\nabla\bar{\epsilon}(x^*)](\bar{x} - x^*) \tag{53}$$

$$= [\nabla\bar{\epsilon}(x^*)^T v]^T(\bar{x} - x^*) = (\lambda v)^T(\bar{x} - x^*) \tag{54}$$

$$= \lambda u \tag{55}$$

Thus, we obtain the test equation $u' = \lambda u$.

# E   TOY ODE PROBLEM

This section aims to demonstrate how the solutions yielded by numerical methods can diverge when the stability regions of the methods are too small. Additionally, we illustrate how our momentum-based techniques can enlarge the stability region. The demonstration is conducted on a 2D toy ODE problem given by:

$$\frac{dx}{dt} = \begin{bmatrix} 0 & 1 \\ -9 & -10 \end{bmatrix} \begin{bmatrix} x_1 \\ x_2 \end{bmatrix}, \qquad x(0) = \begin{bmatrix} -1 \\ 0 \end{bmatrix}. \tag{56}$$

The eigenvalues of the $2 \times 2$ matrix are $-9$ and $-1$, and the exact solution of Equation 56 is given by:

$$x(t) = \frac{1}{8} \begin{bmatrix} 1 \\ -9 \end{bmatrix} e^{-9t} + \frac{9}{8} \begin{bmatrix} -1 \\ 1 \end{bmatrix} e^{-t}. \tag{57}$$

As $t$ increases, $x(t)$ converges to the origin.

Let us say we want to numerically compute $x(3)$ by integrating the ODE for 26 steps with a numerical method. We divide the time interval $[0, 3]$ into 26 equal intervals, resulting in a step size of $\delta = 3/26$. For this particular setting, it turns out that the $2^{\text{nd}}$-order Adams-Bashforth (AB2) method diverges, but the Euler method converges. To see this, observe that the stability region of the AB2 method only cover the interval $[-1, 0]$ of the real line, as depicted in Figures 14b and 14d. So, for the eigenvalue $\lambda = -9$, the product $\delta\lambda = -27/26$ lies just outside the region. Consequently, the numerical solution yielded by AB2 diverges, as indicated by the blue line in Figures 14a and 14c. In contrast, the Euler method's stability region contains both values of $\delta\lambda$, and the numerical solution, represented by the green line in Figures 14a and 14c, appears to be more accurate.

We can improve the stability from AB2 method by applying any of our proposed techniques: Heavy Ball momentum (HB) and GHVB. The stability regions of the modified AB2 methods are given by the red ($\beta = 0.8$) and yellow ($\beta = 0.9$) lines in Figures 14b and 14d. Observe that they contains the points associated with the $\lambda\delta$ values. As a result, the numerical solutions of the modified methods converge to the origin, as demonstrated by the red and yellow lines in Figures 14a and 14c, respectively.

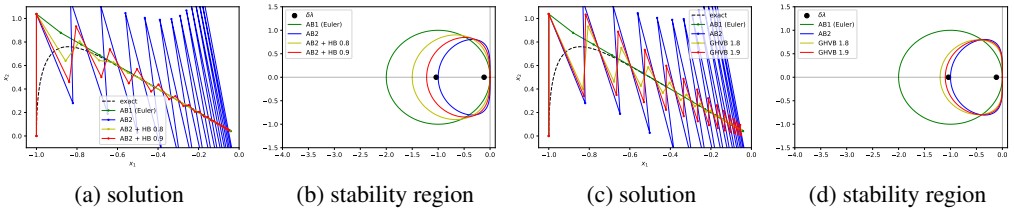

| (a) solution | (b) stability region | (c) solution | (d) stability region |

Figure 14: Comparison of solution trajectories and stability regions of various numerical methods when applied to the toy ODE problem. Here, we seek to compute $x(3)$ in 26 steps with the Euler method, the AB2 method, and methods resulting from modifying AB2 with our momentum-based techniques. Subfigure (a) presents the numerical solutions obtained using our modified AB2 method with HB momentum, while subfigure (c) showcases those obtained using our GHVB. The stability regions of the methods are depicted in subfigures (b) and (d) respectively.

# F   IMPLEMENTATION DETAILS OF PLMS WITH HB AND GHVB METHODS

In this section, we present the complete algorithms for the PLMS method with the HB momentum and the GHVB method in Algorithm 1 and Algorithm 2, respectively.

# G   VARIANCE MOMENTUM METHODS

In 2018, a variant of Polyak's HB momentum called aggregated momentum (Lucas et al., 2018) was proposed. Its objective is to enhance stability while also offering convergence advantages.

---

**Algorithm 1:** PLMS step with HB momentum

---

**input:** $\bar{x}_n$ (previous result), $\delta$ (step size),
$\{e_i\}_{i<n}$ (evaluation buffer), $r$ (method order),
$v_n$ (previous velocity) ;
    $e_n = \bar{\epsilon}_\sigma(\bar{x}_n)$ ;
    $c = \min(r, n)$ ;
    **if** $c == 1$ **then**
        $\hat{e} = e_n$ ;
    **else if** $c == 2$ **then**
        $\hat{e} = (3e_n - e_{n-1})/2$ ;
    **else if** $c == 3$ **then**
        $\hat{e} = (23e_n - 16e_{n-1} + 5e_{n-2})/12$ ;
    **else**
        $\hat{e} = (55e_n - 59e_{n-1} + 37e_{n-3} - 9e_{n-4})/24$ ;
    $v_{n+1} = (1 - \beta)v_n + \beta\hat{e}$;
    **Result:** $\bar{x}_n + \delta v_{n+1}$

---

**Algorithm 2:** GHVB step

---

**input:** $\bar{x}_n$ (previous result), $\delta$ (step size), $\beta$ (damping parameter)
$\{v_i\}_{i\leq n}$ (evaluation buffer), $r$ (method order), ;
    $v_{n+1} = (1 - \beta)v_n + \beta\bar{\epsilon}_\sigma(\bar{x}_n)$ ;
    $c = \min(r, n)$ ;
    **if** $c == 1$ **then**
        $\hat{e} = v_{n+1}$ ;
    **else if** $c == 2$ **then**
        $\hat{e} = ((2 + \beta)v_{n+1} - (2 - \beta)v_n)/2\beta$ ;
    **else if** $c == 3$ **then**
        $\hat{e} = ((18 + 5\beta)v_{n+1} - (24 - 8\beta)v_n$
          $+(6 - \beta)v_{n-1})/12\beta$ ;
    **else if** $c == 4$ **then**
        $\hat{e} = ((46 + 9\beta)v_{n+1} - (78 - 19\beta)v_n$
          $+(42 - 5\beta)v_{n-1} - (10 - \beta)v_{n-2})/24\beta$ ;
    **else**
        $\hat{e} = ((1650 + 251\beta)v_{n+1} - (3420 - 646\beta)v_n$
          $+(2880 - 264\beta)v_{n-1} - (1380 - 106\beta)v_{n-2}$
          $+(270 - 19\beta)v_{n-3})/720\beta$ ;
    **Result:** $\bar{x}_n + \delta\hat{e}$

---

This modification introduces multiple velocities, denoted by $v_n^{(i)}$, each associated with its specific damping coefficient $\beta^{(i)}$.

$$v_{n+1}^{(i)} = (1 - \beta^{(i)})v_n^{(i)} + \beta^{(i)}f(x_n), \qquad x_{n+1} = x_n + \delta\sum_{i=1}^{K} w^{(i)}v_{n+1}^{(i)} \tag{58}$$

Nesterov's momentum (Nesterov, 1983) is one version of the classic momentum that can also be applied to diffusion sampling processes to improve stability. It can be written as follows:

$$y_{n+1} = x_n + \delta\beta f(x_n), \qquad x_{n+1} = y_{n+1} + (1 - \beta)(y_{n+1} - y_n) \tag{59}$$

In fact, Nesterov's momentum can be obtained from aggregated momentum by considering the following:

$$v_{n+1}^{(1)} = (1 - \beta)v_n^{(1)} + \beta f(x_n), \qquad v_{n+1}^{(2)} = f(x_n),$$
$$x_{n+1} = x_n + \delta((1 - \beta)v_{n+1}^{(1)} + \beta v_{n+1}^{(2)}). \tag{60}$$

The stability regions of Nesterov's momentum when applied to the Euler method and high-order Adams-Bashforth methods are illustrated in Figures 15a through 15d. Observe that the stability

regions of methods with Nesterov's momentum become larger in a similar manner to those with Polyak's HB momentum. However, the enlargement due to Nesterov's momentum is more pronounced in the vertical direction, while the Polyak's HB momentum's enlargement is more horizontal in nature. (See Figures 14b and 14d) The differences in the shapes of the stability regions suggest one type of momentum is more suitable to certain ODE problems than the other.

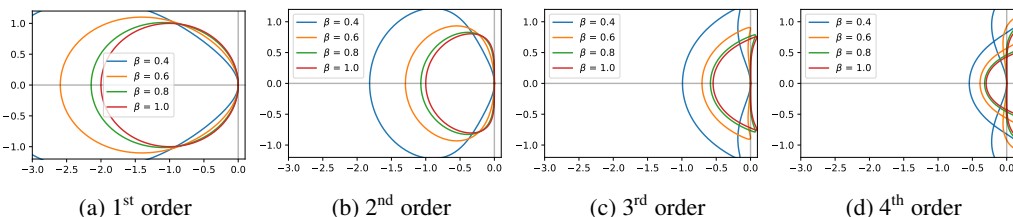

(a) $1^{\text{st}}$ order     (b) $2^{\text{nd}}$ order     (c) $3^{\text{rd}}$ order     (d) $4^{\text{th}}$ order

Figure 15: Comparison of stability regions for different methods with different levels of Nesterov's momentum.

While generalizing the aggregated momentum method to higher-order methods is possible, it is no longer as straightforward as it is with the HB method. As an example, we will consider the $2^{\text{nd}}$-order generalization of Nesterov's momentum method.

We begin by noting that $(\beta + (1 - \beta)\Delta)v_{n+1} = \beta f(x_n)$. Our goal is to find polynomials $B$ and $C$ such that

$$\Delta x_{n+1} = \delta(B(\Delta)v_{n+1} + C(\Delta)f(x_n)). \tag{61}$$

Multiplying both sides of the equation by $(\beta + (1 - \beta)\Delta)$, we get

$$(\beta + (1 - \beta)\Delta)\Delta x_{n+1} = \delta(\beta B(\Delta)f(x_n) + (\beta + (1 - \beta)\Delta)C(\Delta)f(x_n)). \tag{62}$$

Replace the left side with the first two terms from Equation 13, we obtain

$$\delta\left(\beta + \frac{2 + \beta}{2}\Delta\right)f(x_n) = \delta(\beta B(\Delta) + (\beta + (1 - \beta)\Delta)C(\Delta))f(x_n). \tag{63}$$

Let $B(\Delta) = b_0 + b_1\Delta$ and $C(\Delta) = c_0$. Then, by balancing the coefficients of $\Delta$ on both sides, we have $1 = b_0 + c_0$ and $\frac{2+\beta}{2} = \beta b_1 + (1 - \beta)c_0$. We can now write the final formulation as follows:

$$x_{n+1} = x_n + \delta(b_0 v_{n+1} + b_1(v_{n+1} - v_n) + c_0 f(x_n)). \tag{64}$$

This suggests that there are countless different ways to expand the stability region of a numerical method, which offer many new research opportunities.

## H    ORDER OF CONVERGENCE

When solving ODEs numerically, it is important to consider the accuracy of the method used. One way to measure accuracy is by considering the order method's of convergence of the. Suppose we have a numerical method of the form

$$A(E)x_n = \delta B(E)f(x_n), \tag{65}$$

where $A(E) = a_0 + a_1 E + a_2 E^2 + ... + a_s E^s$ and $B(E) = b_0 + b_1 E + ... + b_s E^s$. The method is said to be of $p^{th}$ order if and only if, for all sufficiently smooth functions $x$, we have that

$$\sum_{m=0}^{s} a_m x(\sigma - m\delta) - \delta \sum_{m=0}^{s} b_m x'(\sigma - m\delta) = \mathcal{O}(\delta^{p+1}), \tag{66}$$

where $x'$ denotes the derivative of $x$.

To derive the order of convergence, we use Taylor expansion for both $x$ and $x'$, yielding

$$
\begin{aligned}
\text{L.H.S.} &= \sum_{m=0}^{s} a_m \sum_{k=0}^{\infty} \frac{(-m\delta)^k}{k!} x^{(k)}(\sigma) - \delta \sum_{m=0}^{s} b_m \sum_{k=0}^{\infty} \frac{(-m\delta)^k}{k!} x^{(k+1)}(\sigma) \\
&= \sum_{k=0}^{\infty} \left( \sum_{m=0}^{s} a_m \frac{(-m\delta)^k}{k!} \right) x^{(k)}(\sigma) - \delta \sum_{k=0}^{\infty} \left( \sum_{m=0}^{s} b_m \frac{(-m\delta)^k}{k!} \right) x^{(k+1)}(\sigma) \\
&= \sum_{k=0}^{\infty} \left( \sum_{m=0}^{s} a_m \frac{(-m\delta)^k}{k!} \right) x^{(k)}(\sigma) + \sum_{k=1}^{\infty} \left( \sum_{m=0}^{s} b_m \frac{m^{k-1}(-\delta)^k}{(k-1)!} \right) x^k(\sigma) \\
&= \sum_{m=0}^{s} a_m + \sum_{k=1}^{\infty} \left( \sum_{m=0}^{s} a_m \frac{m^k}{k!} + \sum_{m=0}^{s} b_m \frac{m^{k-1}}{(k-1)!} \right) x^k(\sigma)(-\delta)^k
\end{aligned}
$$

where $x^{(k)}(\sigma)$ denotes the $k^{\text{th}}$ derivative of $x$ evaluated at $\sigma$.

Therefore, the method has $p^{\text{th}}$ order of convergence if and only if the coefficients satisfy the conditions given by

$$
\sum_{m=0}^{s} a_m = 0,
$$

$$
\sum_{m=0}^{s} a_m \frac{m^k}{k!} + \sum_{m=0}^{s} b_m \frac{m^{k-1}}{(k-1)!} = 0, \quad k = 0, 1, ..., p. \tag{67}
$$

Now, we discuss the convergence order of any numerical method after HB momentum is applied to it. An example of such an algorithm is the modified PLMS method, presented in Algorithm 1.

**Theorem 1** (Convergence order of numerical methods with HB momentum). *Suppose that a $p^{th}$-order numerical method has the form $x_{n+1} = x_n + \delta \sum_{m=0}^{s} b_m f(x_{n-m})$, where $p \geq 1$. The modified method that uses HB momentum can be expressed as follows:*

$$
v_{n+1} = (1 - \beta)v_n + \beta \sum_{m=0}^{s} b_m f(x_{n-m}), \tag{68}
$$

$$
x_{n+1} = x_n + \delta v_{n+1}. \tag{69}
$$

*It has first-order convergence.*

*Proof.* From the condition given in 67, it follows that $\sum_{m=0}^{s} b_m = 1$. We can rewrite these equations as:

$$
x_{n+1} - x_n - (1 - \beta)(x_n - x_{n-1}) = \delta\beta \sum_{m=0}^{s} b_m f(x_{n-m}). \tag{70}
$$

To estimate the order of the modified method, we evaluate Equation 70 and obtain:

$$
\sum_{m=0}^{s} a_m = 1 - 1 - (1 - \beta)(1 - 1) = 0, \tag{71}
$$

$$
\sum_{m=0}^{s} a_m \frac{m^1}{1!} + \sum_{m=0}^{s} b_m \frac{m^0}{0!} = 0 - 1 - (1 - \beta)(1 - 2) + \beta \sum_{m=0}^{s} b_m
$$

$$
= -\beta + \beta = 0. \tag{72}
$$

Therefore, we have shown that the modification to the method has first-order convergence. $\qquad \square$

Next, we turn our attention to the GHVB method.

**Theorem 2** (Convergence order of the GHVB method). *The $r^{th}$-order GHVB (Algorithm 2) has order of convergence of $r$.*

*Proof.* We will use the 2$^{\text{nd}}$-order method as an example. Using Equation 16, we can write an equivalent equation as:

$$x_{n+1} - x_n - (1 - \beta)(x_n - x_{n-1}) = \delta \left( \frac{2 + \beta}{2} f(x_n) - \frac{2 - \beta}{2} f(x_{n-1}) \right). \tag{73}$$

To estimate the order of the modified method, we evaluate Equation 70 and obtain:

$$\sum_{m=0}^{s} a_m = 1 - 1 - (1 - \beta)(1 - 1) = 0,$$

$$\sum_{m=0}^{s} a_m \frac{m^1}{1!} + \sum_{m=0}^{s} b_m \frac{m^0}{0!} = 0 - 1 - (1 - \beta)(1 - 2) + \left( \frac{2 + \beta}{2} - \frac{2 - \beta}{2} \right) = 0,$$

$$\sum_{m=0}^{s} a_m \frac{m^2}{2!} + \sum_{m=0}^{s} b_m \frac{m^1}{1!} = \frac{1}{2}(0 - 1^2 - (1 - \beta)(1^2 - 2^2)) + \left( \frac{2 + \beta}{2} 1^1 - \frac{2 - \beta}{2} 2^1 \right)$$

$$= \frac{2 - 3\beta}{2} - \frac{2 - 3\beta}{2} = 0.$$

Thus, the method has a convergence order of two. Methods of other orders can be dealt with in a similar fashion. □

## I  ELABORATION ON THE ORDER OF CONVERGENCE APPROXIMATION

In Appendix H, we explored the theoretical aspects of the order of convergence for numerical methods. In this section, we will delve into the estimation of the order of convergence specifically for GHVB in Section 4.2.

To assess the order of convergence, we focus on the error $e$, referred to as the global truncation error. This error is quantified by measuring the absolute difference in the latent space between the numerical solution and an approximate exact solution obtained through 1,000-step DDIM sampling. The order of convergence for a numerical method is defined as $q$, where the error $e$ follows the relationship $e = \mathcal{O}(\delta^q)$, with $\delta$ representing the step size.

To estimate the order of convergence practically, we adopt a straightforward approach. It involves selecting two distinct step sizes, denoted as $\delta_{\text{new}}$ and $\delta_{\text{old}}$, and computing the corresponding errors $e_{\text{new}}$ and $e_{\text{old}}$. These errors can be approximated using the following formulas:

$$e_{\text{new}} \approx C_{\text{new}}(\delta_{\text{new}})^q, \qquad e_{\text{old}} \approx C_{\text{old}}(\delta_{\text{old}})^q \tag{74}$$

Here, we make the assumption that $C_{\text{new}}$ is approximately equal to $C_{\text{old}}$. By taking the ratio of $e_{\text{new}}$ to $e_{\text{old}}$, we obtain:

$$\frac{e_{\text{new}}}{e_{\text{old}}} \approx \left( \frac{\delta_{\text{new}}}{\delta_{\text{old}}} \right)^q \tag{75}$$

Consequently, we can estimate the order of convergence, denoted as $q$, by evaluating the logarithmic ratio of errors and step sizes:

$$q \approx \frac{\log(e_{\text{new}}/e_{\text{old}})}{\log(\delta_{\text{new}}/\delta_{\text{old}})} \tag{76}$$

In our investigation of GHVB in Section 4.2, we conducted sampling experiments using 20, 40, 80, 160, 320, and 640 steps. This choice of an exponential sequence for the number of steps was intentional, as it allowed us to approximate $\delta_{\text{new}}/\delta_{\text{old}} \approx 1/2$. By doing so, we facilitated the estimation process. The results, representing the approximated order of convergence for GHVB, are visually depicted in Figure 12.

## J  ADDITIONAL QUALITATIVE COMPARISONS ON ARTIFACTS MITIGATION

Figure 1 compares our momentum-based methods, HB and GHVB, with two different diffusion solver methods, DPM-Solver++ (Lu et al., 2022b) and PLMS4 (Liu et al., 2022a), without momentum. The number of sampling steps is held constant while varying the guidance scale $s$ to

intentionally induce divergence artifacts. (Note that the guidance scales that yield such artifacts are different between diffusion models.) The figure demonstrates that, under the difficult settings of low step counts and high guidance scales where the baseline methods produce artifacts, our proposed techniques can successfully eliminate them.

We present additional qualitative results to show the effect of the damping parameter $\beta$ on the quality of images generated by methods modified with HB momentum. We use methods of varying orders, including DPLM-Solver++ (Lu et al., 2022b), UniPC (Zhao et al., 2023), and PLMS4(Liu et al., 2022a). The diffusion models utilized in our analysis are Realistic Vision v2.0 [1], Anything Diffusion v4.0[2], Counterfeit Diffusion V2.5[3], Pastel-Mix[4], Deliberate Diffusion[5], and Dreamlink Diffusion V1.0[6]. The results are shown in Figure 16. Notice that stronger momentum (lower $\beta$) leads to fewer and less severe artifacts.

## K ADDITIONAL EXPERIMENT ON ADM

We present additional details and results for the ADM experiment in Section 5.2. The primary objective of this experiment was to provide a quantitative evaluation of class-conditional diffusion sampling in the context of pixel-based images. The experiment was conducted using the pre-trained diffusion and classifier model at the following link: [7]. The implementation used in our experiment was obtained directly from the official DPM-Solver GitHub repository [8].

To enhance the capabilities of DPM-Solver++, we incorporated HB momentum into DPM-Solver++ (just change a few lines of code) and implemented the splitting method LTSP both with and without HB momentum into the DPM-Solver code for comparative purposes. The experiment was done on four NVIDIA RTX A4000 GPUs and a 24-core AMD Threadripper 3960x CPU.

The results of the experiment, as measured by the full FID score, are presented in Table 1 (as well as in Figure 9). Our technique is highlighted in grey within the table, while the best FID score for each number of steps is indicated in bold. Additionally, Figure 17 showcases sample images from this experiment. As this experiment uses pixel-based diffusion models, the observed divergence artifacts differ from those in latent-based diffusion models. Specifically, the artifacts may display excessive brightness or darkness caused by pixel values nearing the maximum or minimum thresholds.

| Method | Number of Steps | | | | | | | |
| --- | --- | --- | --- | --- | --- | --- | --- | --- |
| | 10 | 12 | 14 | 16 | 18 | 20 | 25 | 30 |
| DPM-Solver++ | 66.77 | 46.77 | 34.56 | 26.97 | 21.87 | 19.48 | 16.31 | 15.63 |
| DPM-Solver++ w/ HB 0.8 | 47.10 | 33.65 | 25.61 | 21.42 | 18.94 | 17.76 | 15.98 | 15.53 |
| LTSP [PLMS4, PLMS1] | 45.32 | 34.08 | 26.58 | 21.54 | 18.54 | 17.15 | **15.79** | **15.51** |
| LTSP [PLMS4, GHVB 0.8] | **37.43** | **29.74** | **23.60** | **20.23** | **18.46** | **17.14** | 16.07 | 15.79 |

Table 1: FID scores on classifier-guidance ADM models

---

[1] https://huggingface.co/SG161222/Realistic_Vision_V2.0
[2] https://huggingface.co/andite/anything-v4.0
[3] https://huggingface.co/gsdf/Counterfeit-V2.5
[4] https://huggingface.co/andite/pastel-mix
[5] https://huggingface.co/XpucT/Deliberate
[6] https://huggingface.co/dreamlike-art/dreamlike-diffusion-1.0
[7] https://github.com/openai/guided-diffusion
[8] https://github.com/LuChengTHU/dpm-solver

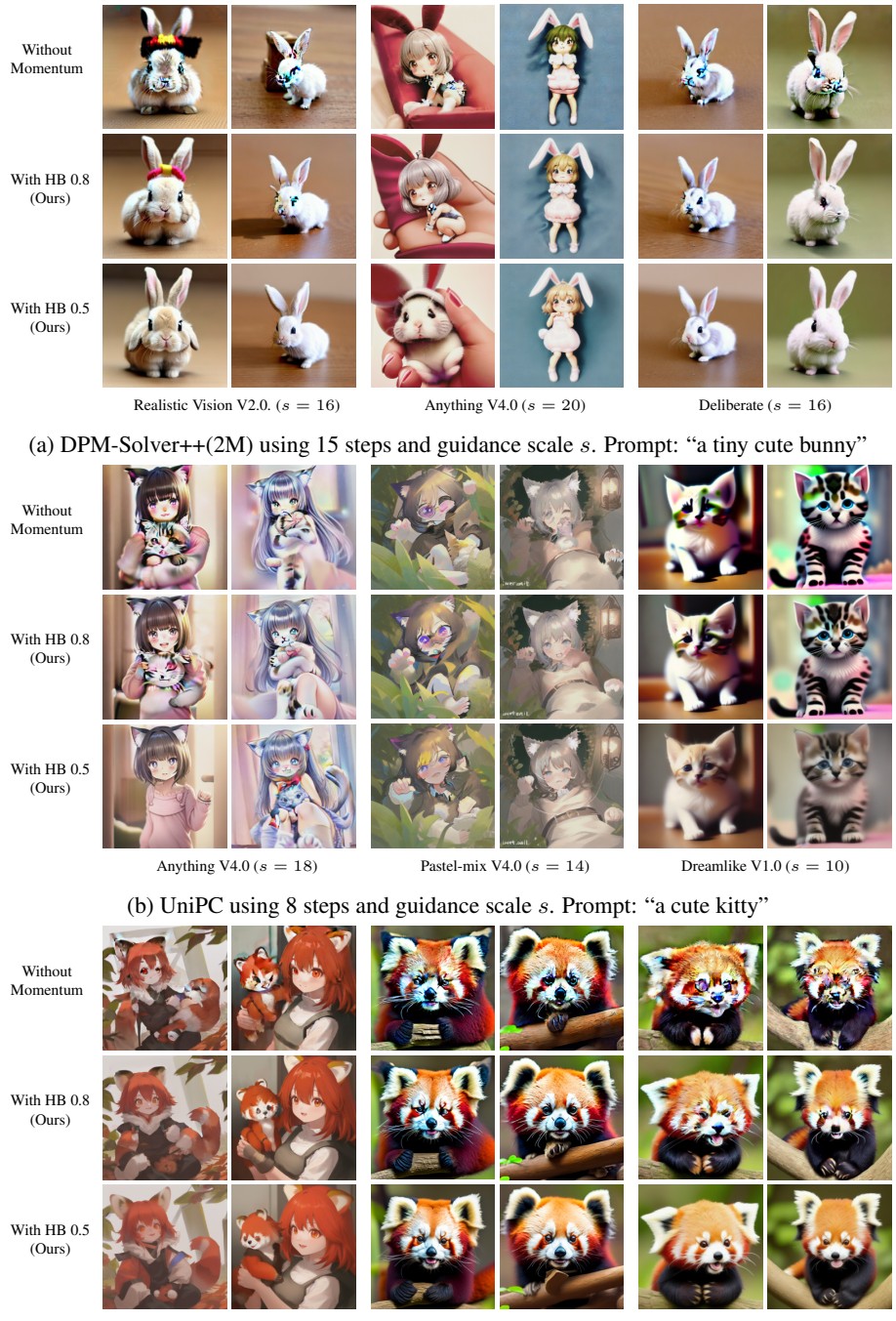

(a) DPM-Solver++(2M) using 15 steps and guidance scale $s$. Prompt: "a tiny cute bunny"

(b) UniPC using 8 steps and guidance scale $s$. Prompt: "a cute kitty"

(c) PLMS4 using 15 steps and guidance scale $s$. Prompt: "cute humanoid red panda"

Figure 16: Impact of different damping coefficients $\beta$ on HB momentum for 2nd-order DPM-Solver++(2M) (Lu et al., 2022b), 3rd-order UniPC (Zhao et al., 2023), and 4th-order (Liu et al., 2022a). Incorporating higher momentum values (lower $\beta$) helps mitigate the occurrence of divergence artifacts.

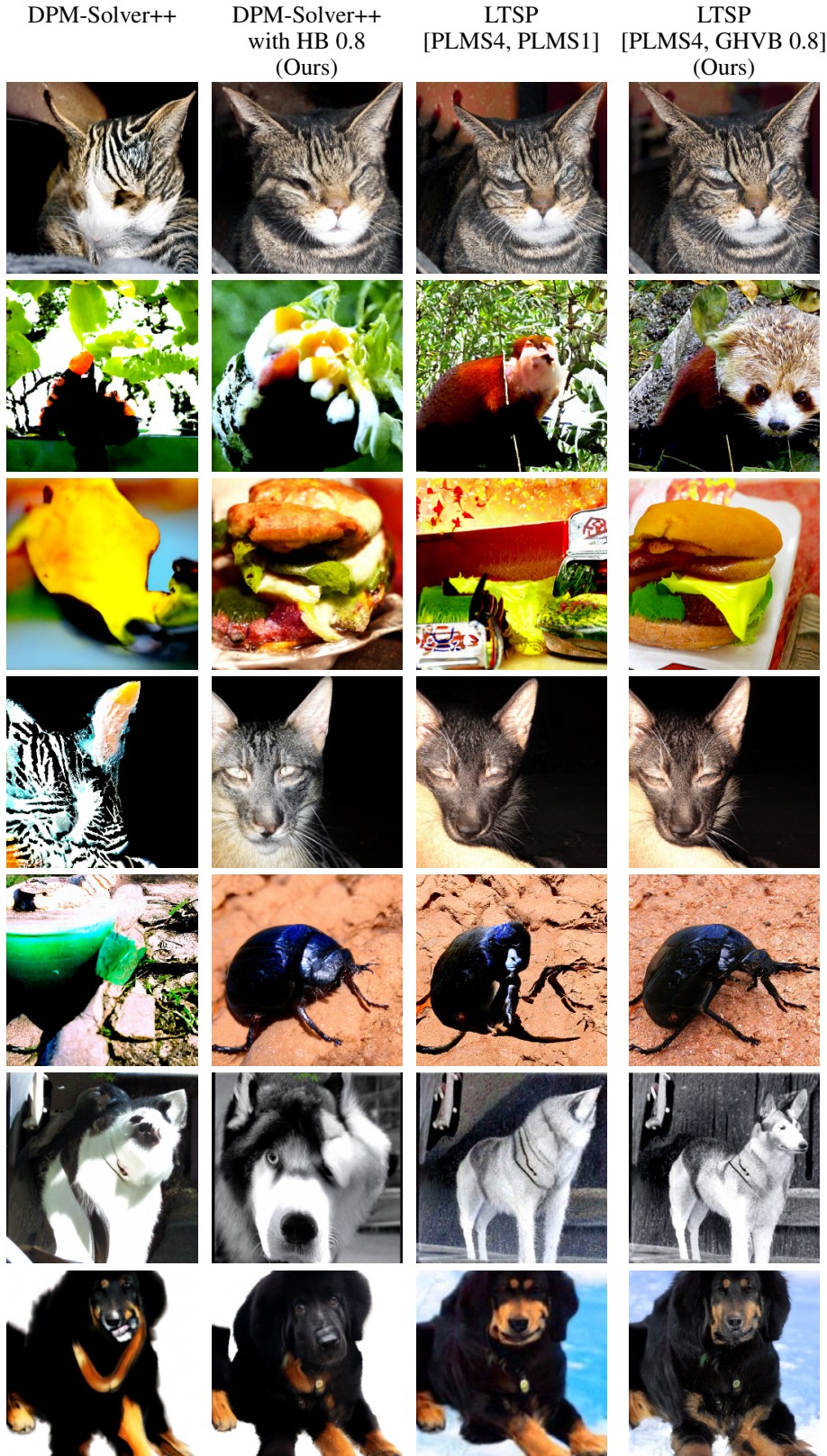

Figure 17: Samples from DPM-Solver++ (Lu et al., 2022b) and LTSP (Wizadwongsa & Suwa-janakorn, 2023) with and without momentum, employing classifier guidance diffusion with a guidance scale of 10 and 20 sampling steps.

## L    ADDITIONAL EXPERIMENT ON DiT

This section supplements the experiment on DiT in Section 5.3 and reports results for two additional baselines: DPM-Solver++ (Lu et al., 2022b) and LTSP4 (Wizadwongsa & Suwajanakorn, 2023). The code implementation and pre-trained DiT model were obtained directly from the official GitHub repository[9]. The experiment was done on four NVIDIA GeForce RTX 2080 Ti GPUs and a 24-core AMD Threadripper 3960x CPU.

Table 2 presents the full results including those of the original baselines. Our technique is highlighted in grey in the table, and the best FID score for each number of steps is in bold. We also include the improved Precision and Recall metrics (Kynkäänniemi et al., 2019) in Tables 3 and 4, respectively, where higher values indicate superior performance. Generated samples for different sampling methods are shown in Figure 18. Our methods produce comparable FID scores to LTSP4 and PLMS4 at 25 steps, but outperform all baselines for lower 6-20 steps.

| Method | Number of Steps | | | | | | | |
|---|---|---|---|---|---|---|---|---|
|  | 6 | 7 | 8 | 9 | 10 | 15 | 20 | 25 |
| DDIM | 55.35 | 36.97 | 26.06 | 19.47 | 15.02 | 8.04 | 6.52 | 5.94 |
| DPM-Solver++ | 18.60 | 10.80 | 7.93 | 6.72 | 6.13 | 5.49 | 5.30 | 5.24 |
| LTSP4 [PLMS4, PLMS1] | 13.33 | 9.01 | 7.49 | 6.55 | 6.09 | 5.32 | 5.20 | **5.17** |
| PLMS4 | 13.10 | 8.94 | 7.31 | 6.51 | 6.03 | 5.32 | 5.21 | **5.17** |
| PLMS4 w/ HB 0.8 | 14.35 | 9.25 | 7.46 | 6.68 | 6.19 | 5.47 | 5.29 | 5.24 |
| PLMS4 w/ HB 0.9 | 11.66 | 8.16 | 6.69 | 6.21 | **5.78** | 5.29 | 5.19 | 5.17 |
| GHVB 3.8 | **10.99** | **7.93** | **6.63** | **6.19** | 5.80 | 5.31 | 5.22 | 5.18 |
| GHVB 3.9 | 11.67 | 8.29 | 6.83 | 6.30 | 5.87 | 5.31 | 5.22 | 5.18 |

Table 2: FID scores on DiT-XL

| Method | Number of Steps | | | |
|---|---|---|---|---|
|  | 6 | 8 | 10 | 20 |
| DDIM | 0.36 | 0.56 | 0.67 | 0.79 |
| DPM-Solver++ | 0.63 | 0.75 | 0.79 | **0.81** |
| LTSP4 | 0.67 | 0.74 | 0.78 | **0.81** |
| PLMS4 | 0.68 | 0.75 | 0.78 | **0.81** |
| PLMS4 w/ HB 0.8 | 0.68 | **0.77** | **0.79** | 0.81 |
| PLMS4 w/ HB 0.9 | 0.70 | **0.77** | **0.79** | 0.81 |
| GHVB3.8 | **0.71** | **0.77** | **0.79** | 0.81 |
| GHVB3.9 | 0.70 | 0.76 | 0.78 | **0.81** |

Table 3: Precision on DiT-XL

| Method | Number of Steps | | | |
|---|---|---|---|---|
|  | 6 | 8 | 10 | 20 |
| DDIM | 0.60 | 0.64 | 0.65 | 0.67 |
| DPM-Solver++ | 0.67 | 0.68 | 0.68 | **0.68** |
| LTSP4 | **0.70** | **0.70** | **0.69** | **0.68** |
| PLMS4 | **0.70** | **0.70** | **0.69** | **0.68** |
| PLMS4 w/ HB 0.8 | 0.68 | 0.68 | 0.67 | **0.68** |
| PLMS4 w/ HB 0.9 | 0.69 | 0.69 | 0.67 | **0.68** |
| GHVB3.8 | 0.69 | **0.70** | 0.68 | **0.68** |
| GHVB3.9 | **0.70** | **0.70** | **0.69** | **0.68** |

Table 4: Recall on DiT-XL

## M    ADDITIONAL EXPERIMENT ON TEXT-TO-IMAGE GENERATION

To provide a more comprehensive evaluation of the methods discussed in Section 5.1, we utilize a fine-tuned variant of Stable-Diffusion called Anything V4. We consider full-path samples generated by PLMS4 (Liu et al., 2022a) at 1,000 steps as reference solutions. The performance of each method is evaluated by measuring the image similarity between the generated samples produced using a reduced number of steps and the reference samples. Importantly, both sets of samples originate from identical initial noise maps. This comparison allows us to assess how well the solution from each configuration matches the full-path reference solution.

---

[9] https://github.com/facebookresearch/DiT

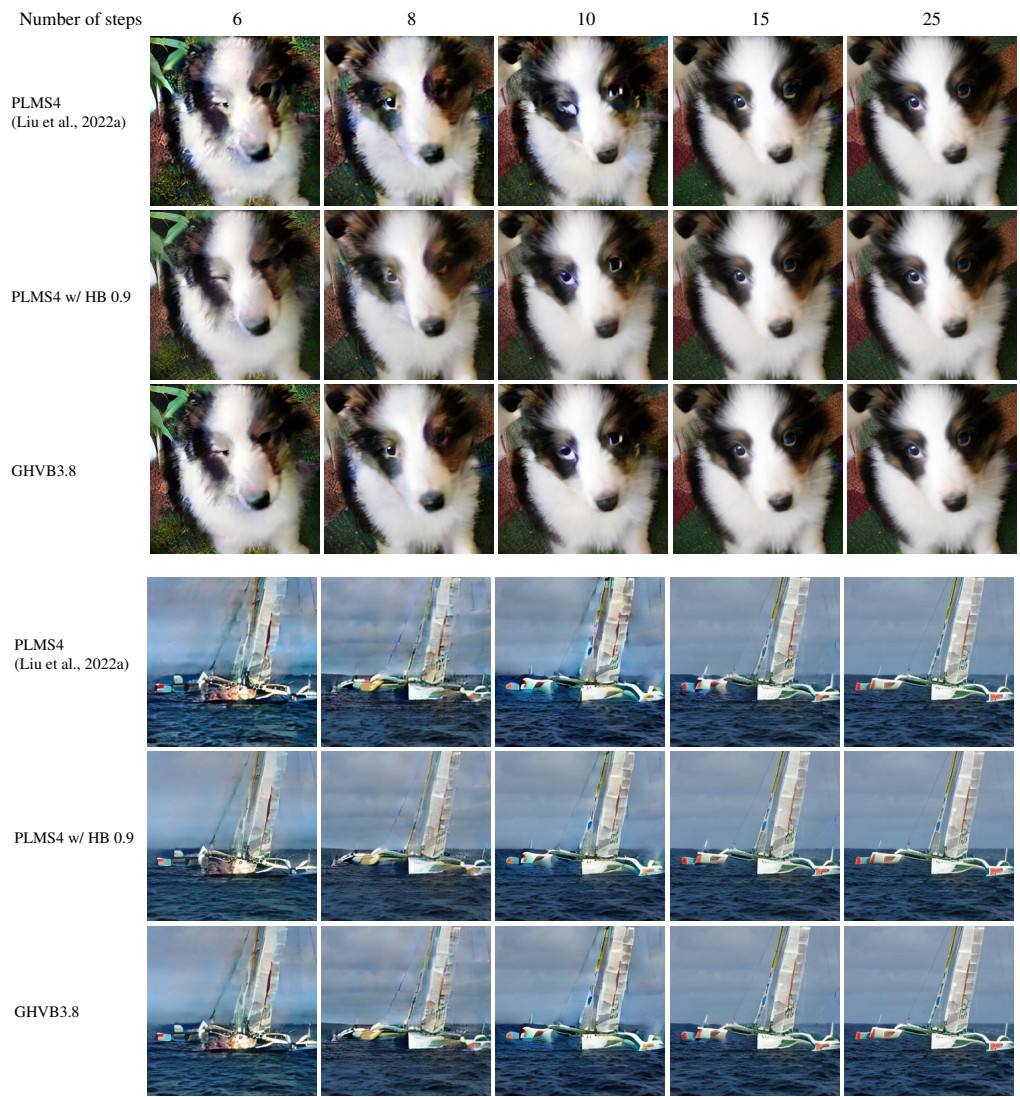

Figure 18: Comparison of samples generated from DiT-XL with a guidance scale of 3, using different sampling methods and sampling steps.

We measure image similarity using Learned Perceptual Image Patch Similarity (LPIPS) (Zhang et al., 2018) and the L2 norm in the latent space, as discussed in Section 4.2. The outcomes of these analyses are visually presented in Figure 19. We also include the results from other metrics, e.g., Structural Similarity Index (SSIM) (Wang et al., 2004) and High-Frequency Error Norm (HFEN) (Ravishankar & Bresler, 2010), in Figure 20.

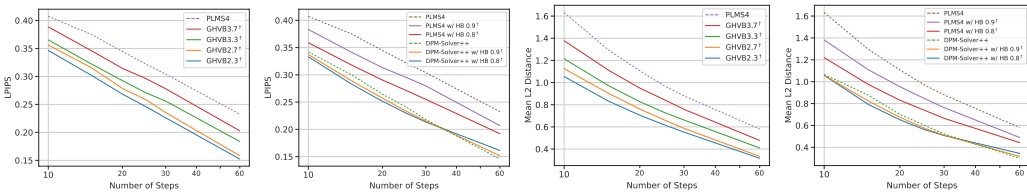

Figure 19: Comparison of LPIPS and L2 distance across different sampling methods, with and without using our momentum techniques. The experimental setting is similar to that in Section 5.1.

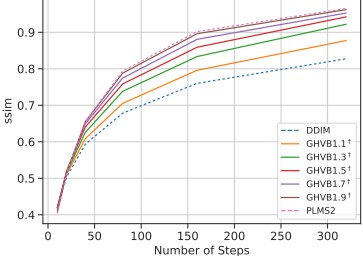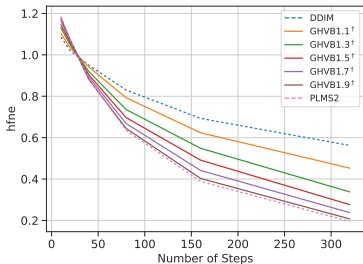

Figure 20: Comparison of SSIM (higher is better) and HFNE (lower is better) distance across different sampling methods, with and without using our momentum techniques. The experimental setting is similar to that in Figure 11.

Discrepancies between the numerical solutions and the 1,000-step reference solution can arise from two primary factors: the accuracy of the employed method and the presence of divergence artifacts. Notably, in this particular context, divergence artifacts tend to outweigh errors stemming from method accuracy. Consequently, higher-order methods exhibit greater discrepancies in both LPIPS and L2 similarity measurements. It is worth highlighting that our techniques demonstrate a remarkable ability to minimize deviations from the reference solution in the majority of cases. Furthermore, Figure 19 consistently demonstrates the superiority of our techniques compared to other methods, as evidenced by the lower LPIPS and L2 similarity scores. These results indicate that our techniques effectively both reduce divergence artifacts and handle errors related to method accuracy.

# N  EXPERIMENT ON MDT

In this section, we conduct an additional experiment using Masked Diffusion Transformer (MDT) Gao et al. (2023) for classifier-free guidance diffusion sampling. The code implementation and MDT model pre-trained on ImageNet256 (Russakovsky et al., 2015) were obtained directly from the official GitHub repository[10]. The experiment was done on two NVIDIA GeForce RTX 3090 GPUs. Similar to the experiment on DiT-XL (Peebles & Xie, 2022) in Section 5.3, we compare FID scores of numerical solvers with our proposed momentum techniques to the baseline of PLMS4 (Liu et al., 2022a) and DDIM (Zhang et al., 2023) using 50,000 generated samples.

As presented in Table 5, both HB and GHVB momentum can mitigate artifacts and lead to improved FID scores compared to PLMS4. Note that using HB 0.8 yields worse FID scores compared to using PLMS4 alone because of its 1$^{st}$-order convergence property. Similar to Appendix L, we also include the improved Precision and Recall metrics (Kynkäänniemi et al., 2019) in Tables 6 and 7. This experiment further highlights that our methods work across various recent diffusion models.

| Method | Number of Steps | | | | | | | |
| | 6 | 7 | 8 | 9 | 10 | 15 | 20 | 25 |
|---|---|---|---|---|---|---|---|---|
| DDIM | 56.53 | 36.79 | 24.17 | 17.13 | 12.62 | 5.11 | 3.40 | 2.77 |
| PLMS4 | 11.10 | 6.44 | 5.45 | 4.30 | 3.89 | 2.53 | 2.15 | 2.01 |
| PLMS4 w/ HB 0.8 | 13.36 | 7.19 | 5.03 | 4.03 | 3.50 | 2.42 | 2.12 | 2.03 |
| PLMS4 w/ HB 0.9 | 10.48 | 5.77 | **4.38** | **3.65** | **3.26** | **2.28** | **2.03** | **1.96** |
| GHVB 3.8 | **9.67** | **5.60** | 4.48 | 3.80 | 3.39 | 2.36 | 2.07 | 1.98 |
| GHVB 3.9 | 10.18 | 5.88 | 4.83 | 3.98 | 3.58 | 2.43 | 2.10 | 1.99 |

Table 5: FID scores on MDT

---

[10]https://github.com/sail-sg/MDT

| Method | Number of Steps | | | |
|---|---|---|---|---|
| | 6 | 8 | 10 | 20 |
| DDIM | 0.33 | 0.53 | 0.64 | 0.76 |
| PLMS4 | 0.66 | 0.71 | 0.73 | 0.76 |
| PLMS4 w/ HB 0.8 | 0.64 | 0.72 | **0.74** | **0.77** |
| PLMS4 w/ HB 0.9 | 0.66 | **0.73** | **0.74** | **0.77** |
| GHVB3.8 | **0.67** | 0.72 | **0.74** | **0.77** |
| GHVB3.9 | **0.67** | 0.71 | 0.73 | **0.77** |

Table 6: Precision on MDT

| Method | Number of Steps | | | |
|---|---|---|---|---|
| | 6 | 8 | 10 | 20 |
| DDIM | 0.50 | 0.54 | 0.57 | 0.63 |
| PLMS4 | **0.62** | **0.65** | **0.66** | **0.66** |
| PLMS4 w/ HB 0.8 | 0.59 | 0.63 | 0.64 | 0.64 |
| PLMS4 w/ HB 0.9 | 0.61 | 0.64 | 0.65 | 0.65 |
| GHVB3.8 | **0.62** | **0.65** | **0.66** | 0.65 |
| GHVB3.9 | **0.62** | **0.65** | **0.66** | **0.66** |

Table 7: Recall on MDT

## O  EXPERIMENT ON PERSONALIZED IMAGE GENERATION

This section evaluates our methods on *personalized image generation*, a task which involves generating images with specific concepts, styles, or following certain compositions given by reference images. The sampling process in this task is significantly more challenging to speed up due to a greater incidence of artifacts compared to the regular diffusion sampling conditioned with only a text prompt. All experiments were conducted on four NVIDIA RTX A4000 GPUs.

### O.1  FACE-IDENTITY PRESERVING IMAGE GENERATION

This experiment focuses on generating images of people's faces with certain face orientations, specified by face key points. We fine-tuned Stable Diffusion (Rombach et al., 2022) using 6 rank-1 LoRAs (Hu et al., 2021) across six facial identities, which was implemented in the GitHub repository[11]. We used a face dataset accessible at Kaggle[12]. For each identity in the dataset, the corresponding LoRA was trained with 10 randomly selected images using *Pivotal Tuning* (Roich et al., 2022), consisting of Textual Inversion (Gal et al., 2022) followed by LoRA DreamBooth (Ruiz et al., 2022) training, for 1,000 steps. Note that we employed Stable Diffusion 1.5[13] as the base model. Our proposed HB methods were applied to PLMS4 (Liu et al., 2022a) and DPM-Solver++ (Lu et al., 2022b), which also served as baselines for our comparison.

At test time, we utilized a variant of Stable Diffusion named DreamShaper[14] and applied the 6 LoRAs with fixed LoRA scaling parameters $\alpha = 1.0$ and $\alpha = 0.4$ for the text encoder and the UNet model, respectively, resulting in 6 different models. The sampling process is further conditioned with the outputs from the ControlNet V1.1[15] trained on a human pose estimation task. Subsequently, we prepared 25 sets of inputs, i.e., text prompt, face key points, random seed, and a fixed guidance scale of 15, and generated 25 converged, reference results by applying 1,000-step PLMS4 sampling to each of the 6 augmented models, resulting in a total of 150 target images. For evaluation, we calculated the mean LPIPS (Zhang et al., 2018) and L2 distance between these images latent and those generated by each ODE solver when provided with the same inputs.

For each numerical solver employing momentum number $\beta$, we conducted a grid search to find the optimal value in the range $[0.1, \ldots, 0.9]$ that yields the least number of sampling steps such that the LPIPS is comparable to that of 200-step DDIM Zhang et al. (2023). As shown in Figure 21, the results indicate that, with the same number of sampling steps, PLMS4 with HB and GHVB generated higher-quality images than vanilla PLMS4. This implies that those with momentum can achieve the target image quality with fewer steps. In particular, our two methods reached the LPIPS threshold set by the 200-step DDIM method with 16.53% and 25.38% less sampling time, respectively, compared to PLMS4 without momentum. A similar result can be observed when comparing DPM-Solver++ and its HB momentum variant, resulting in 18.22% less sampling time. Figures 23 and 24 provide examples of the generated images.

---

[11]https://github.com/cloneofsimo/lora
[12]https://www.kaggle.com/datasets/vishesh1412/celebrity-face-image-dataset
[13]https://huggingface.co/runwayml/stable-diffusion-v1-5
[14]https://huggingface.co/Lykon/DreamShaper
[15]https://huggingface.co/lllyasviel/control_v11p_sd15_openpose

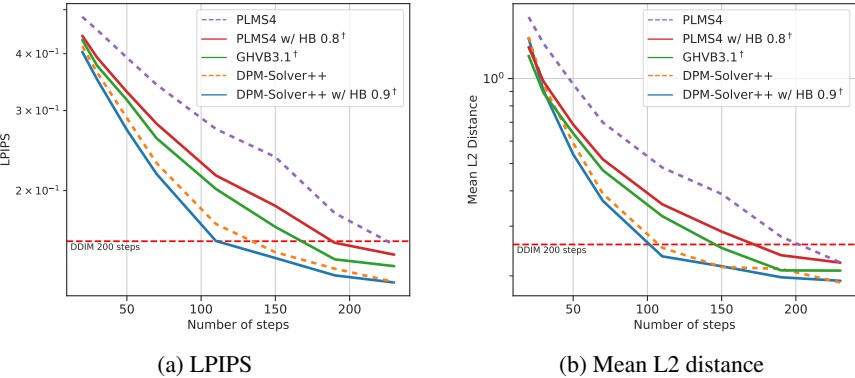

(a) LPIPS         (b) Mean L2 distance

Figure 21: Comparison of LPIPS and L2 distance of (a) PLMS4 with and without momentum, (b) DPM-Solver++ with and without momentum, and (c) GHVB in the face-identity preserving image generation experiment.

## O.2 REFERENCE-ONLY CONTROL GENERATION

We evaluate our techniques on the *reference-only control pipeline* implemented for Stable Diffusion in the following GitHub repository[16]. This pipeline conditions the sampling process on a text prompt and a reference image, enabling content and style transfer of the reference image via attention mechanism and feature rescaling proposed in AdaIN (Huang & Belongie, 2017).

We utilize a fine-tuned Stable Diffusion model named Anything V3.0[17] to generate Japanese animation-style images. Specifically, we prepared 150 sets of inputs, i.e., text prompt, reference image, random seed, and a fixed guidance scale of 7.5 and style fidelity of 1.0 (see [16]), and generated the target results by using a 1,000-step PLMS4 (Liu et al., 2022a). For evaluation, we calculated the mean LPIPS (Zhang et al., 2018) and L2 distance between the psudo-converged result and those generated by numerical solvers when provided with the same inputs.

The results in Figure 22 suggest that PLMS4 with HB and GHVB generated higher-quality images than the regular PLMS4. Specifically, our two methods reached the LPIPS threshold set by the 150-step DDIM method with 32.40% and 42.92% less sampling time, respectively, compared to PLMS4 without momentum. Figure 25 provides examples of the generated images.

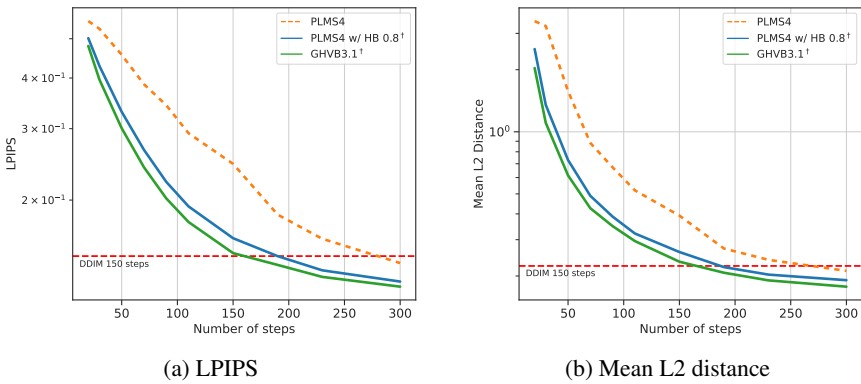

(a) LPIPS         (b) Mean L2 distance

Figure 22: Comparison of LPIPS and L2 distance of (a) PLMS4 with and without momentum and (b) GHVB in the reference-only control experiment.

---

[16] https://github.com/huggingface/diffusers/blob/main/examples/community/stable_diffusion_reference.py

[17] https://huggingface.co/Linaqruf/anything-v3.0

[18] https://huggingface.co/datasets/yuvalkirstain/beautiful_interesting_spectacular_photo_anime_25000

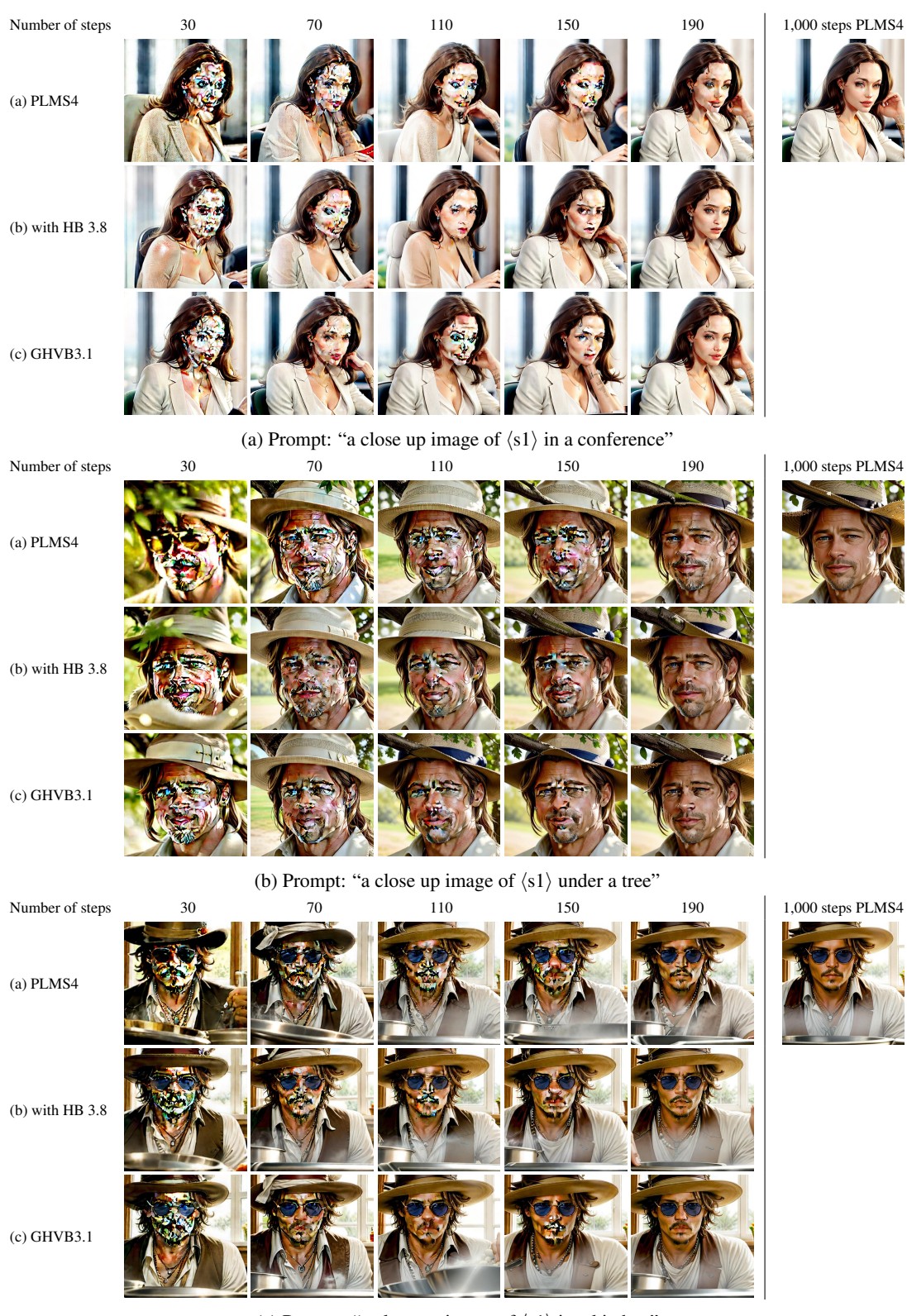

(a) Prompt: "a close up image of ⟨s1⟩ in a conference"

(b) Prompt: "a close up image of ⟨s1⟩ under a tree"

(c) Prompt: "a close up image of ⟨s1⟩ in a kitchen"

Figure 23: Comparison of samples generated from a fine-tuned Stable Diffusion model called DreamShaper with (a) PLMS4 (Liu et al., 2022a), (b) PLMS4 with HB 0.8, and (c) GHVB3.1 using different number of sampling steps and guidance scale $s = 15$.

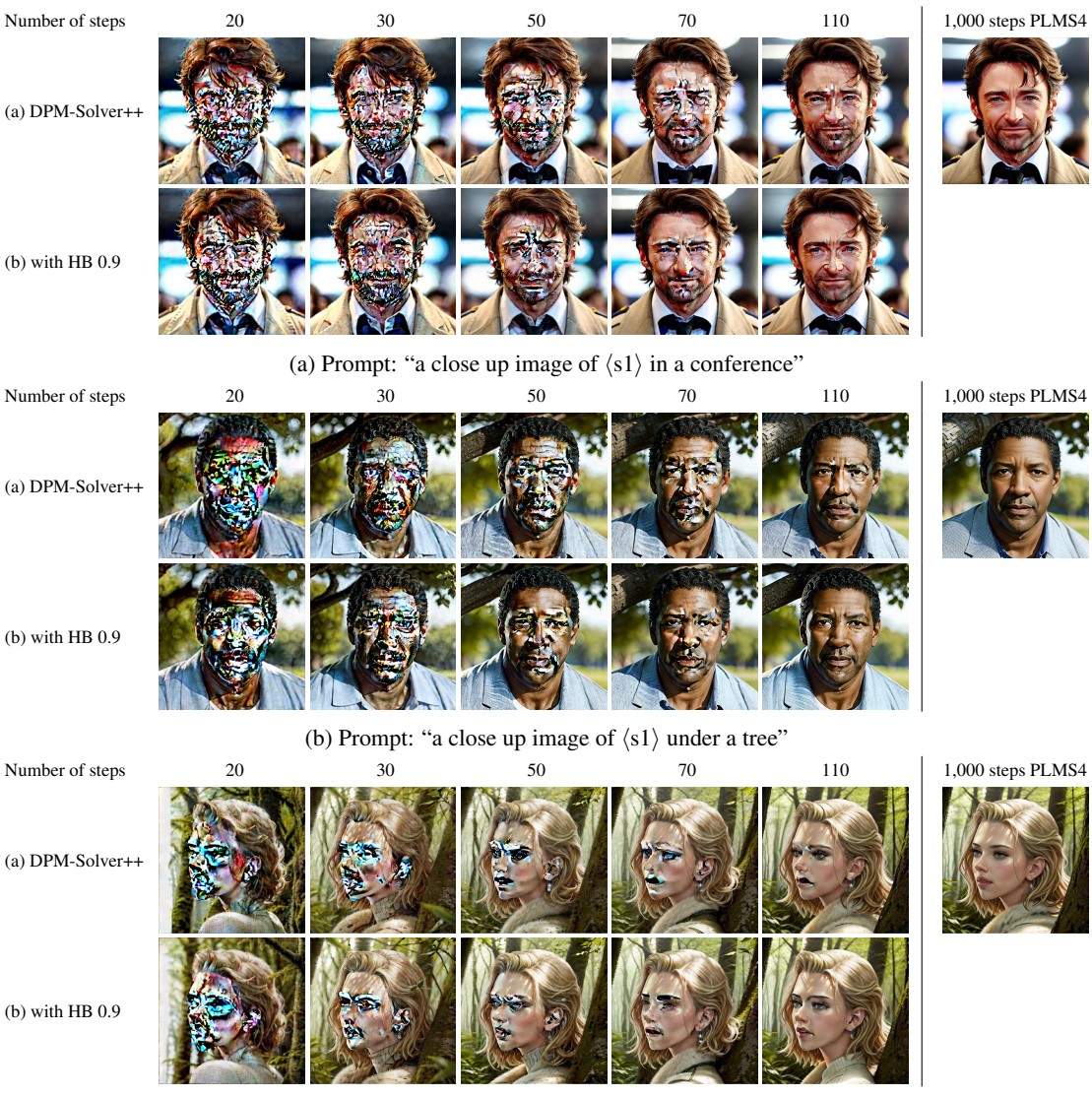

Figure 24: Comparison of samples generated from a fine-tuned Stable Diffusion model called DreamShaper with (a) DPM-Solver++ (Lu et al., 2022b) and (b) DPM-Solver++ with HB 0.9 using different number of sampling steps and guidance scale $s = 15$.

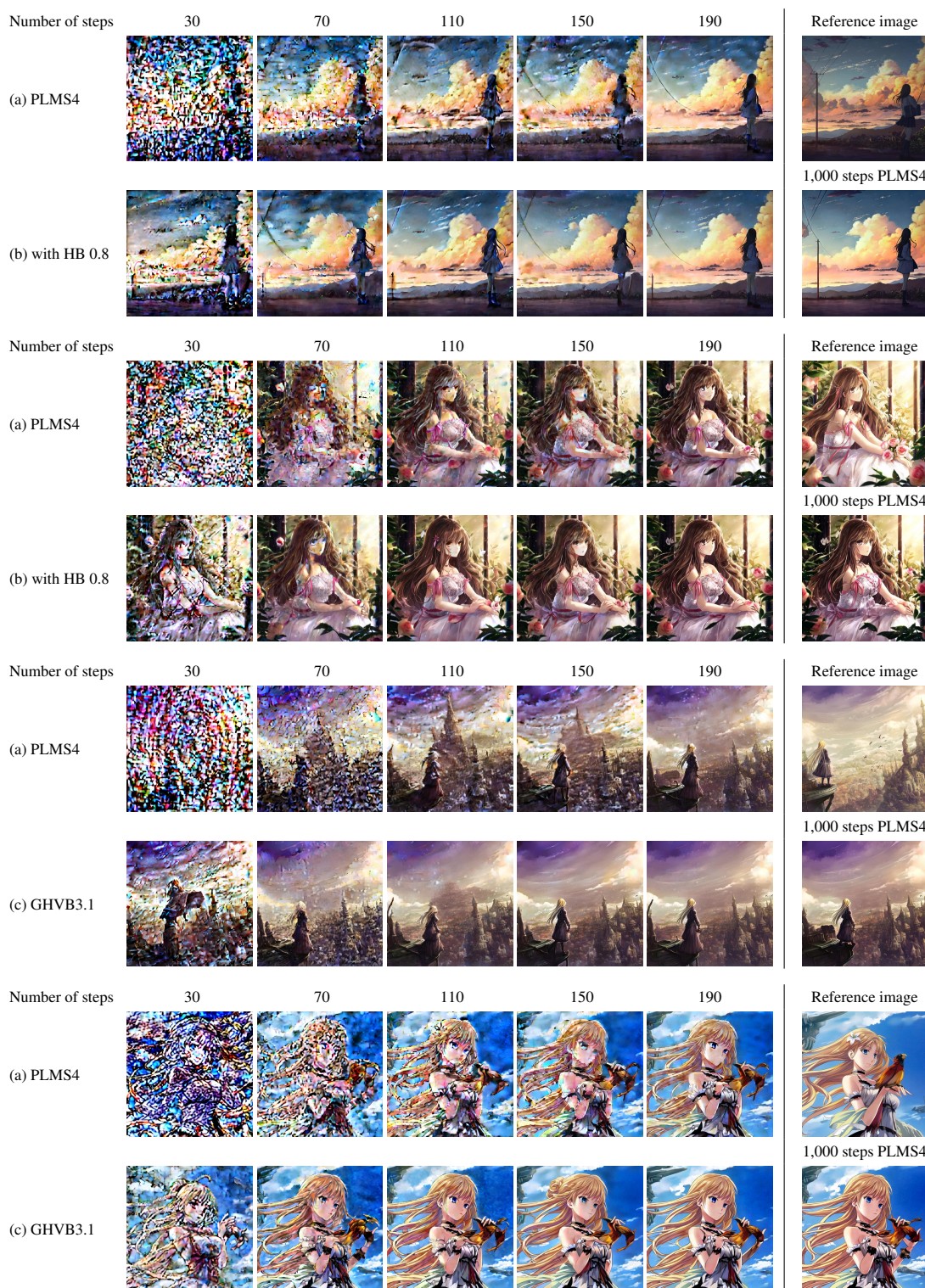

Figure 25: Comparison of samples generated from a fine-tuned Stable Diffusion model named Anything V3.0 in reference-only control pipeline with (a) PLMS4 (Liu et al., 2022a), (b) PLMS4 with HB 0.8, and (c) GHVB3.1 using different number of sampling steps and guidance scale $s = 12$. Reference images are available on Hugging Face[18]. Prompt: "1girl"

# P  FACTORS CONTRIBUTING TO ARTIFACT OCCURRENCE IN FINE-TUNED DIFFUSION MODELS

We investigate three factors that influence the occurrence of divergence artifacts in diffusion sampling: number of steps, guidance scale, and choice of diffusion models. We present results obtained from Stable Diffusion 1.5 (Figure 27) and other models fine-tuned to generate images in specific styles: Midjourney (Figure 29), Japanese animation (Figure 31), and photorealistic (Figure 33).

Our observations reveal that insufficient numbers of steps and high guidance scales positively correlate with the presence of divergence artifacts in the generated samples. Fine-tuned models exhibit a higher sensitivity to these factors, resulting in a greater incidence of artifacts compared to Stable Diffusion 1.5. Consistent with the findings presented in Section 3.2, reducing the number of steps increases the likelihood of artifact occurrence. Furthermore, increasing the guidance scale amplifies the magnitude of eigenvalues, contributing to the presence of artifacts.

The choice of diffusion model also impacts artifact occurrence. Stable Diffusion 1.5 produces the fewest artifacts compared to the fine-tuned models. Among these fine-tuned models, Openjourney, which yields results similar to those obtained from Stable Diffusion 1.5, exhibits the lowest occurrence of artifacts. This suggests that extensive changes to the model may alter the eigenvalues and lead to an increased presence of artifacts.

Additionally, we present the results of our techniques for mitigating divergence artifacts in Figure 28, 30, 32, and 34. The parameter $\beta$ that offers the optimal tradeoff between reducing artifacts and maintaining accuracy depends on the chosen guidance scale and diffusion model.

# Q  ABLATION STUDY ON HB MOMENTUM

This section provides a comprehensive analysis of the convergence speed of Polyak's Heavy Ball (HB) momentum, which can be incorporated into existing diffusion sampling methods by modifying a few lines of code. Specifically, we compare the results obtained from several methods with and without HB momentum with the targets generated from the 1,000-step DDIM, using LPIPS in the image space and L2 in the latent space. We then estimate their orders of convergence, as explained in Appendix I. The results are visually presented in Figure 26.

In contrast to the interpolation-like behavior observed in Figure 12 for GHVB, we observe that the use of HB momentum leads to an increase in both the LPIPS score and the L2 distance when selecting values of $\beta$ that are less than 1. This is even worse than the 1st-order method DDIM when $\beta$ is below 0.7. These findings indicate a deviation from the desired convergence behavior, highlighting a potential decrease in solution accuracy, even though HB momentum has been shown to successfully mitigate divergence artifacts.

Additionally, we find that the numerical orders of convergence also tend to approach the same value. These observations align with our analysis in Theorem 1 of Appendix H, indicating that when $\beta$ deviates from 1, the employed approach exhibits 1st-order convergence and is unable to achieve high-order convergence. These conclusions emphasize the importance of carefully considering the choice of $\beta$ in order to strike a balance between convergence speed and solution quality. Further details and insights into the performance of the HB momentum approach can be obtained from Figure 26, enhancing our understanding of its behavior within the context of the studied problem.

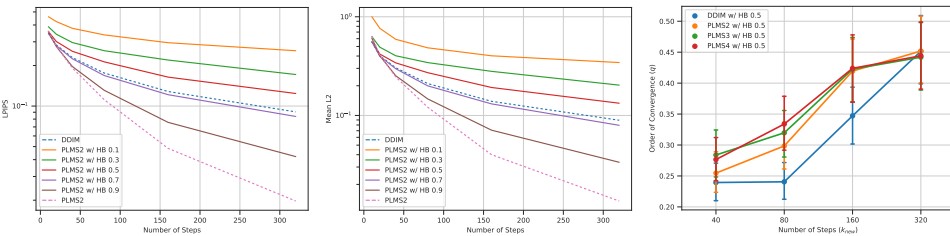

Figure 26: Comparison of LPIPS, mean L2 distance, and order of convergence of HB when using different damping coefficients. Statistical means are averaged from 160 initial latent codes.

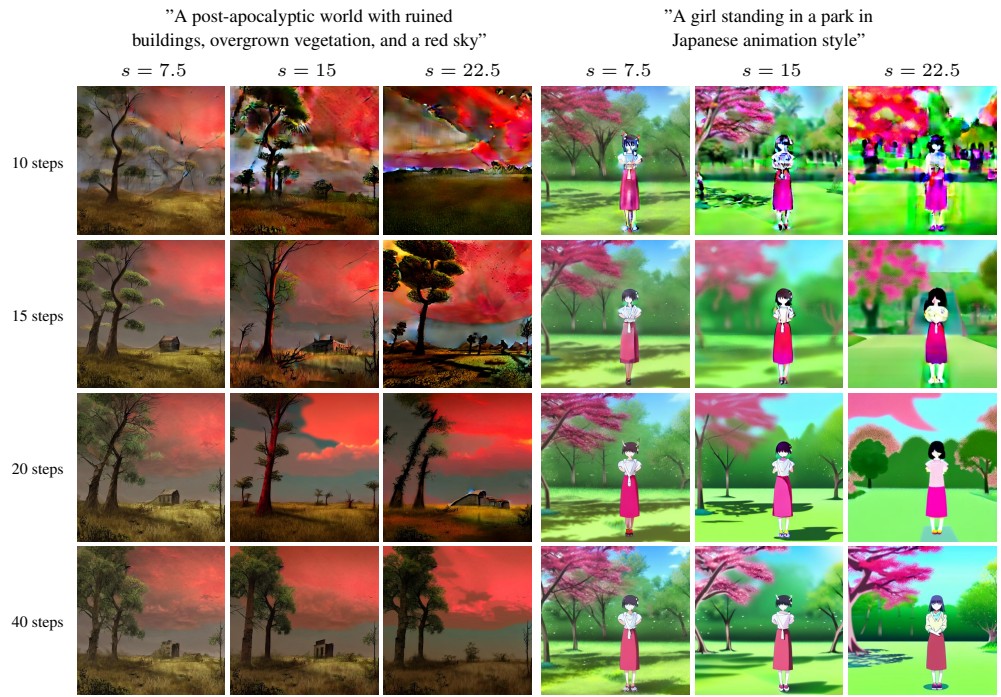

Figure 27: Comparison of samples generated from Stable Diffusion 1.5 [19]using PLMS4 (Liu et al., 2022a) with different sampling steps and guidance scale.

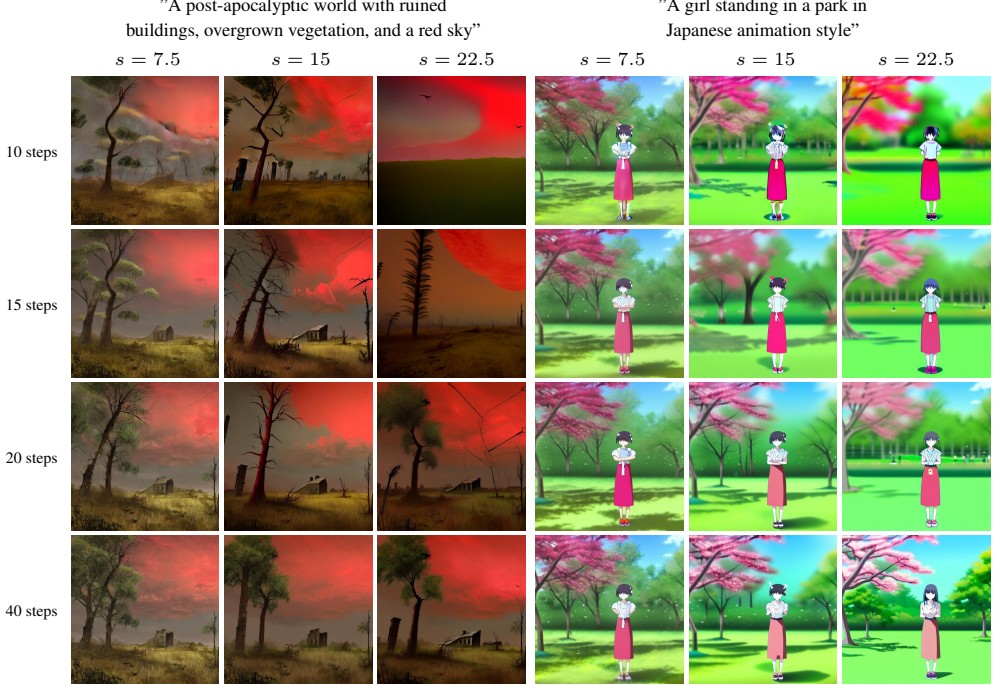

Figure 28: Comparison of samples generated from Stable Diffusion 1.5 using PLMS4 with HB $\beta$ under various sampling steps and guidance scale $s$. Specifically, we employ $\beta = 0.9$ for $s = 7.5$, $\beta = 0.8$ for $s = 15$, and $\beta = 0.6$ for $s = 22.5$ to account for the varying degrees of artifact manifestation associated with each guidance scale.

---

[19]https://huggingface.co/runwayml/stable-diffusion-v1-5

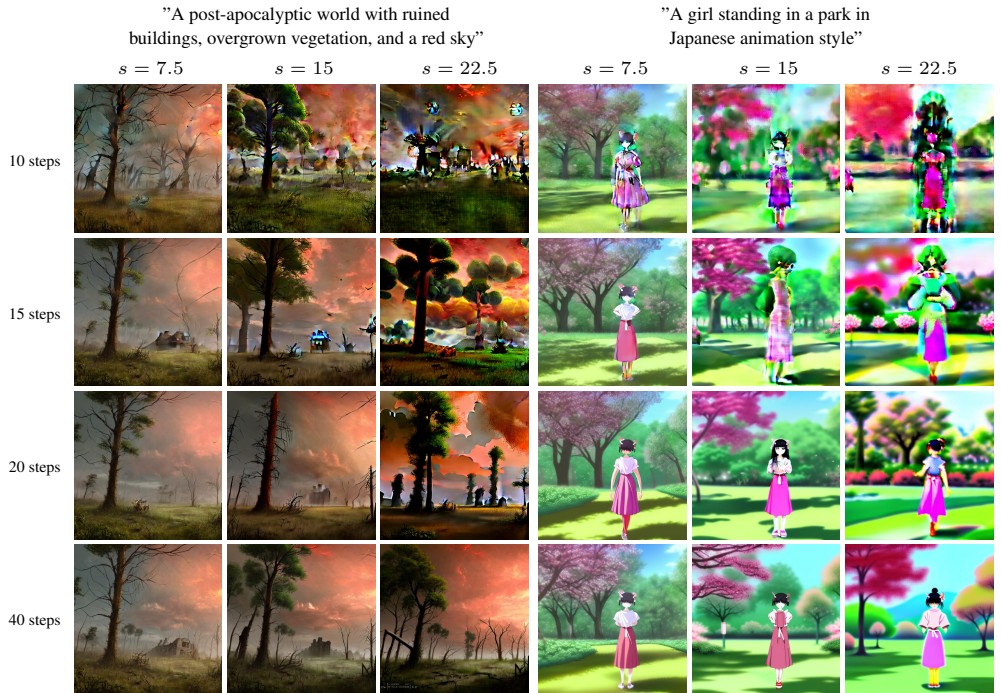

Figure 29: Comparison of samples generated from Openjourney [20]using PLMS4 (Liu et al., 2022a) with different sampling steps and guidance scale.

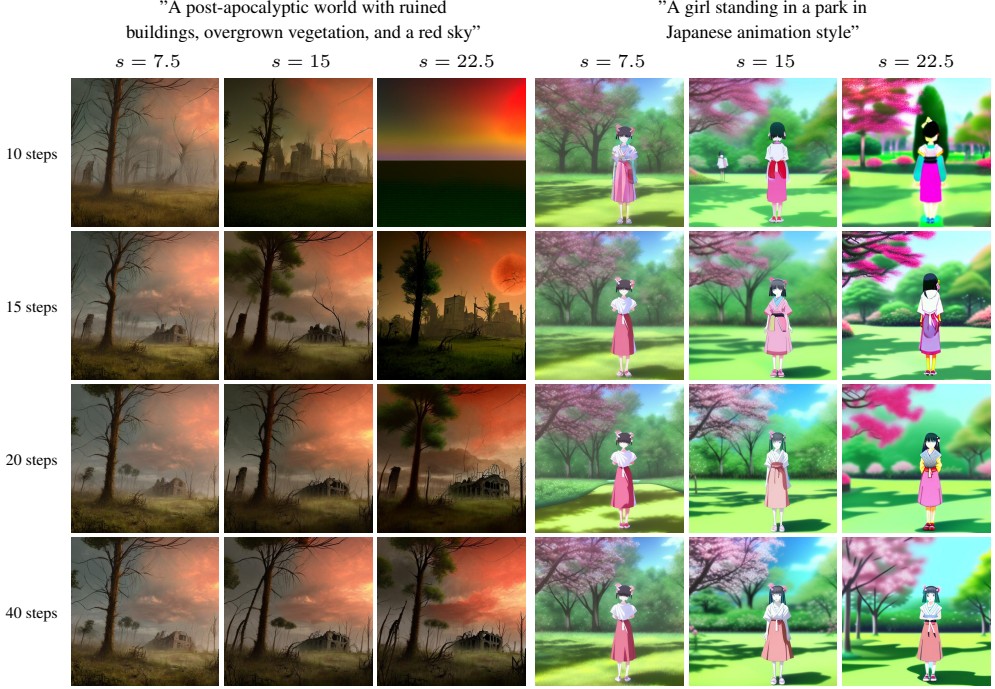

Figure 30: Comparison of samples generated from Openjourney using PLMS4 with HB $\beta$ under various sampling steps and guidance scale $s$. Specifically, we employ $\beta = 0.8$ for $s = 7.5$, $\beta = 0.6$ for $s = 15$, and $\beta = 0.6$ for $s = 22.5$ to account for the varying degrees of artifact manifestation associated with each guidance scale.

---

[20]https://huggingface.co/prompthero/openjourney

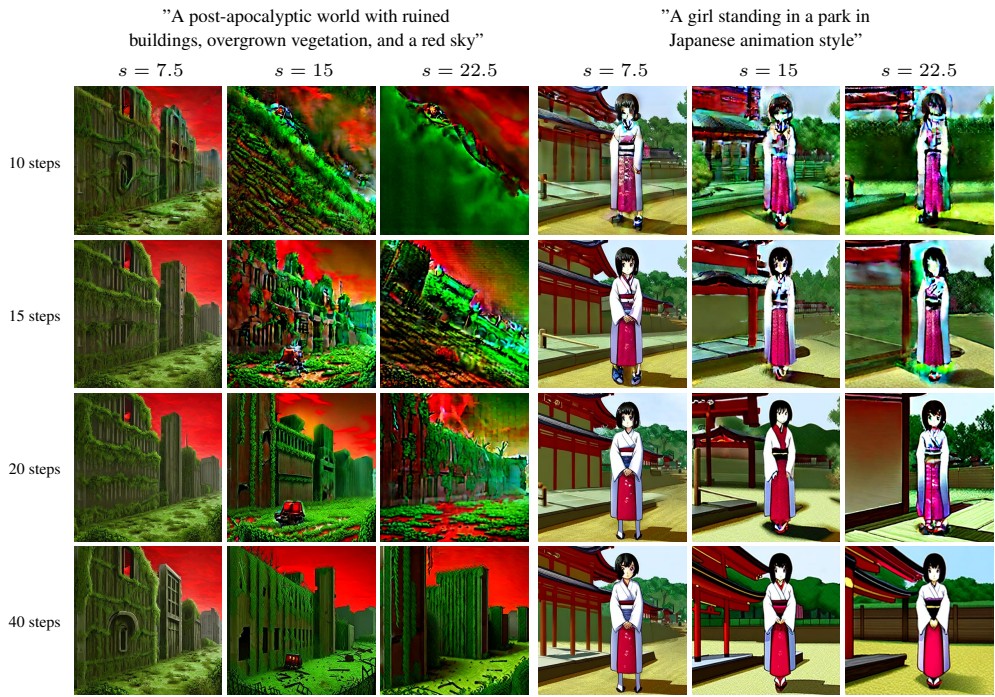

Figure 31: Comparison of samples generated from Waifu Diffusion V1.4 [21] using PLMS4 (Liu et al., 2022a) with different sampling steps and guidance scale.

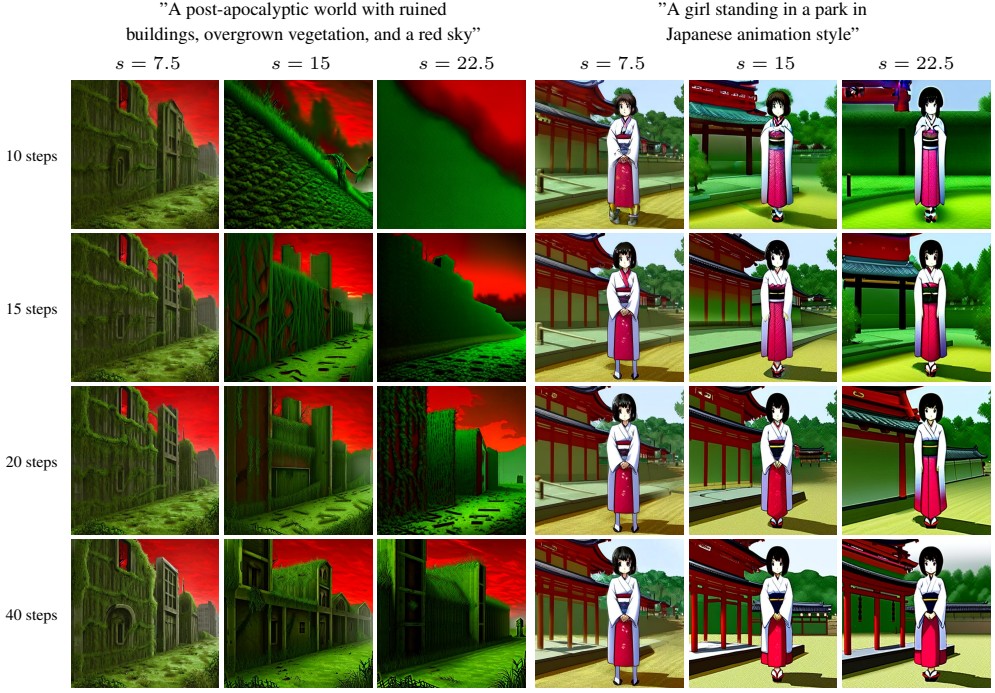

Figure 32: Comparison of samples generated from Waifu Diffusion V1.4 using PLMS4 with HB $\beta$ under various sampling steps and guidance scale $s$. Specifically, we employ $\beta = 0.8$ for $s = 7.5$, $\beta = 0.7$ for $s = 15$, and $\beta = 0.6$ for $s = 22.5$ to account for the varying degrees of artifact manifestation associated with each guidance scale.

---

[21] https://huggingface.co/hakurei/waifu-diffusion

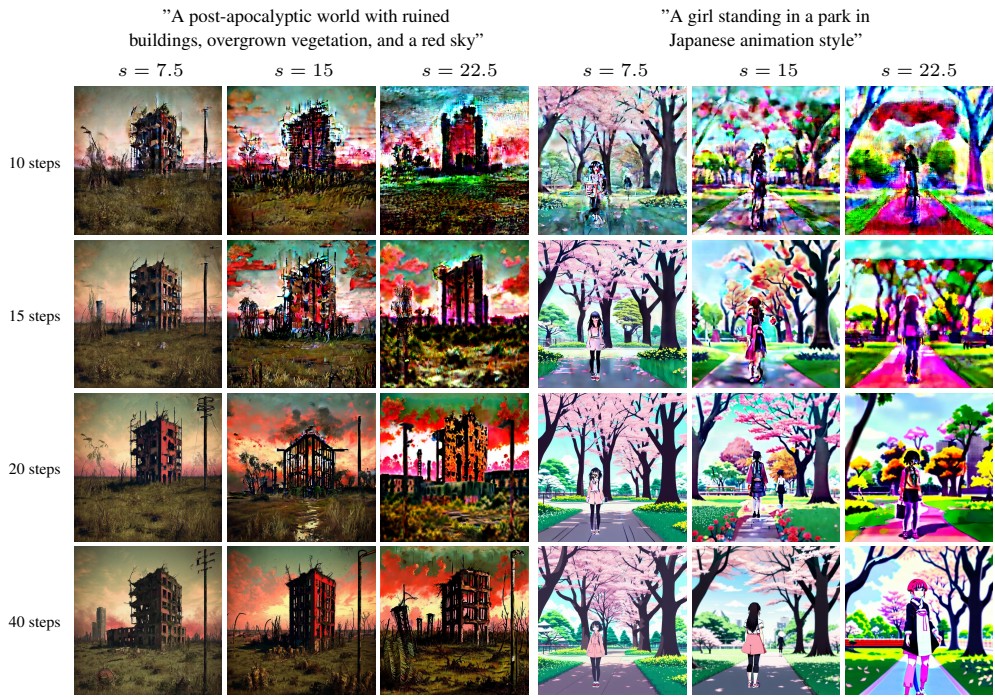

Figure 33: Comparison of samples generated from Dreamlike Photoreal 2.0 [22]using PLMS4 (Liu et al., 2022a) with different sampling steps and guidance scale.

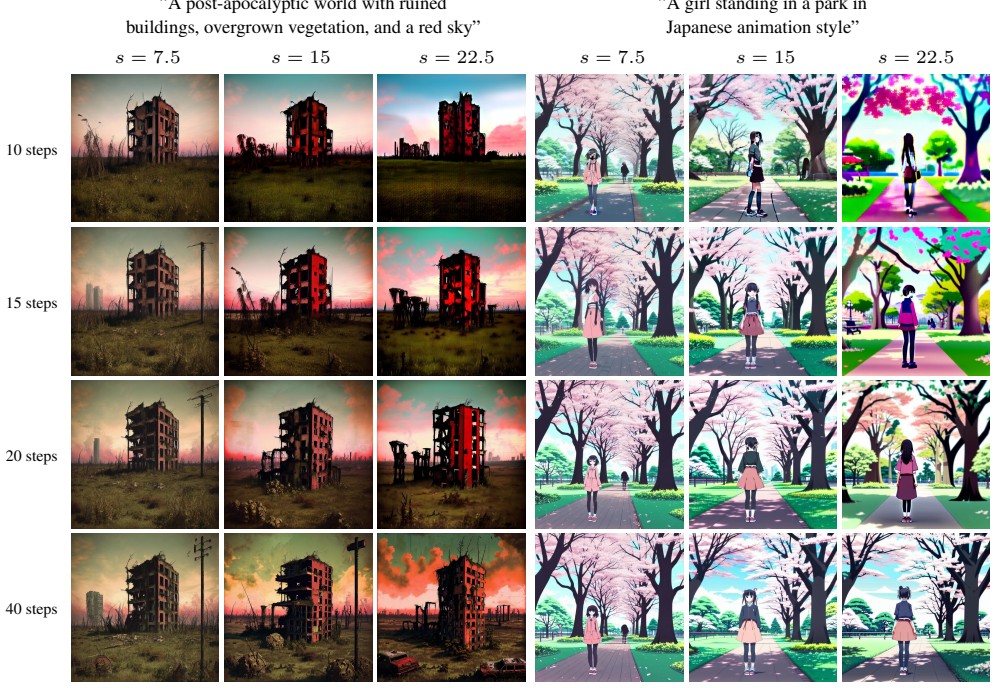

Figure 34: Comparison of samples generated from Dreamlike Photoreal V2.0 using PLMS4 with HB $\beta$ under various sampling steps and guidance scale $s$. Specifically, we employ $\beta = 0.7$ for $s = 7.5$, $\beta = 0.6$ for $s = 15$, and $\beta = 0.6$ for $s = 22.5$ to account for the varying degrees of artifact manifestation associated with each guidance scale.

---

[22]https://huggingface.co/dreamlike-art/dreamlike-photoreal-2.0

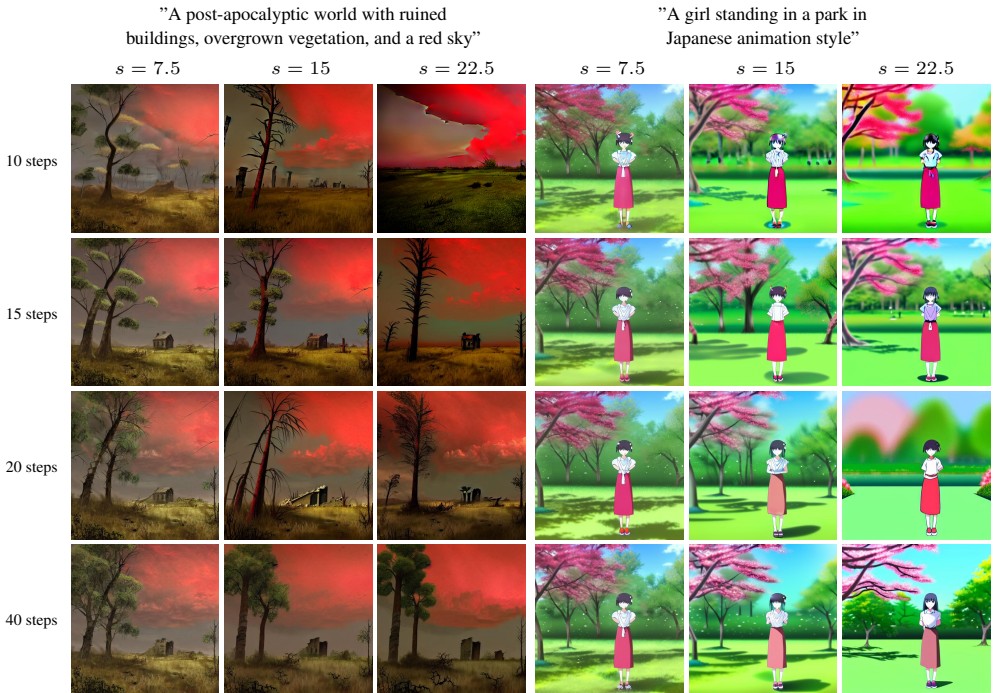

Figure 35: Comparison of samples generated from Stable Diffusion 1.5 using GHVB($3.0 + \beta$) under various sampling steps and guidance scale $s$. Specifically, we employ $\beta = 0.5$ for $s = 7.5$, $\beta = 0.2$ for $s = 15$, and $\beta = 0.1$ for $s = 22.5$ to account for the varying degrees of artifact manifestation associated with each guidance scale.

## R    ABLATION STUDY ON NESTEROV MOMENTUM

In Appendix G, we investigated the potential of incorporating different types of momentum, such as Nesterov's momentum, into existing diffusion sampling methods to mitigate divergence artifacts. Similar to the analysis conducted in Section 5.4 and Appendix Q, the primary objective of this section is to explore the convergence speed of Nesterov's momentum by comparing two key metrics: LPIPS in the image space and L2 in the latent space.

Figure 36 presents the results, which reveal intriguing parallels with the behavior of HB momentum observed in Figure 26. When Nesterov's momentum is applied to the PLMS2 method, the solution accuracy progressively diminishes as the value of $\beta$ deviates from 1, as indicated by the corresponding increase in both LPIPS and L2 metrics. Notably, the model's accuracy drops below that of the DDIM when $\beta$ falls below 0.5. Qualitative comparisons of these two types of momentum are shown in Figure 37.

Furthermore, our analysis of the order of convergence demonstrates that Nesterov's momentum does not achieve a high order of convergence, similar to HB momentum. These findings emphasize the importance of carefully considering the choice of momentum method, along with the specific values assigned to $\beta$, in order to strike an optimal balance between convergence speed and solution quality.

## S    ABLATION STUDY ON MAGNITUDE SCORE

In this section, our objective is to provide further verification and justification of the experiment conducted in Section 5.1 by exploring various parameter settings for the magnitude score and assessing their effects on the selected model.

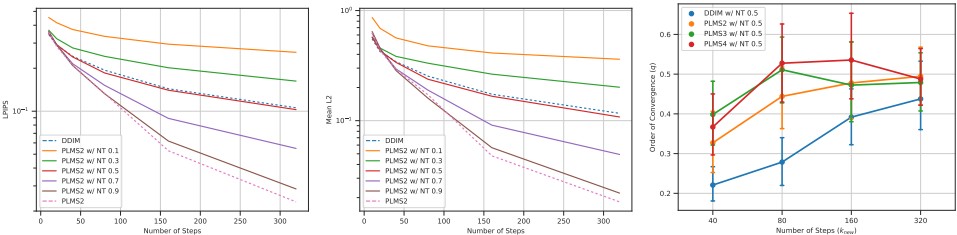

Figure 36: Comparison of LPIPS, mean L2 distance, and order of convergence of Nesterov's momentum when using different damping coefficients. Statistical means are averaged from 160 initial latent codes.

## S.1 Results with Alternative Parameter Settings

To gain deeper insights into the integration of momentum into sampling methods, we analyze the results of the magnitude scores depicted in Figure 7 (Section 5.1). This analysis involves varying the threshold $\tau$ and the kernel size $k$ for max-pooling in the calculation of the magnitude score. By investigating different parameter settings, we aim to validate the outcomes of the experiment and uncover the scaling impact of the magnitude score. The results, shown in Figure 38, highlight the influence of threshold $\tau$ and kernel size $k$ on the magnitude score. It is important to note that while extreme values of $\tau$ or $k$ may introduce ambiguity in interpreting the outcomes, the overall observed trends remain consistent.

## S.2 Results on Alternative Models

In this section, we present the findings from our analysis conducted on alternative diffusion models, namely Stable Diffusion 1.5, Waifu Diffusion V1.4, and Dreamlike Photoreal 2.0. The primary aim of this investigation is to assess the impact of different models on the magnitude score and determine whether the trends identified in Section 5.1 hold across diverse model architectures.

For this analysis, we employed the same magnitude score parameters as in Section 5.1. The results of our examination are illustrated in Figure 39, which showcases the magnitude scores for each model. One important observation is that the change in model architecture only affects the scale of the magnitude score, while the overall trend remains consistent across all models.

## T Statistical Reports

In this section, we present detailed statistical reports for the experiments conducted in Section 5.1 and Section 4.2. These reports provide detailed information, including mean values and their corresponding 95% confidence intervals, to offer a thorough understanding of the experimental results.

Firstly, we focus on the experiment related to mitigating the magnitude score in Section 5.1. The results depicted in Figure 7 are presented in Table 8. Additionally, the outcomes illustrated in Figure 8 are reported in Table 9. For the ablation study of GHVB in Section 4.2, we provide the results shown in Figure 11 in Table 10 and report the findings in Figure 12 in Table 11.

Furthermore, we include a runtime comparison of each sampling method in Table 12, detailing the wall clock time required for each method. The results indicate that all the methods exhibit similar sampling times, ensuring a fair comparison across the different approaches.

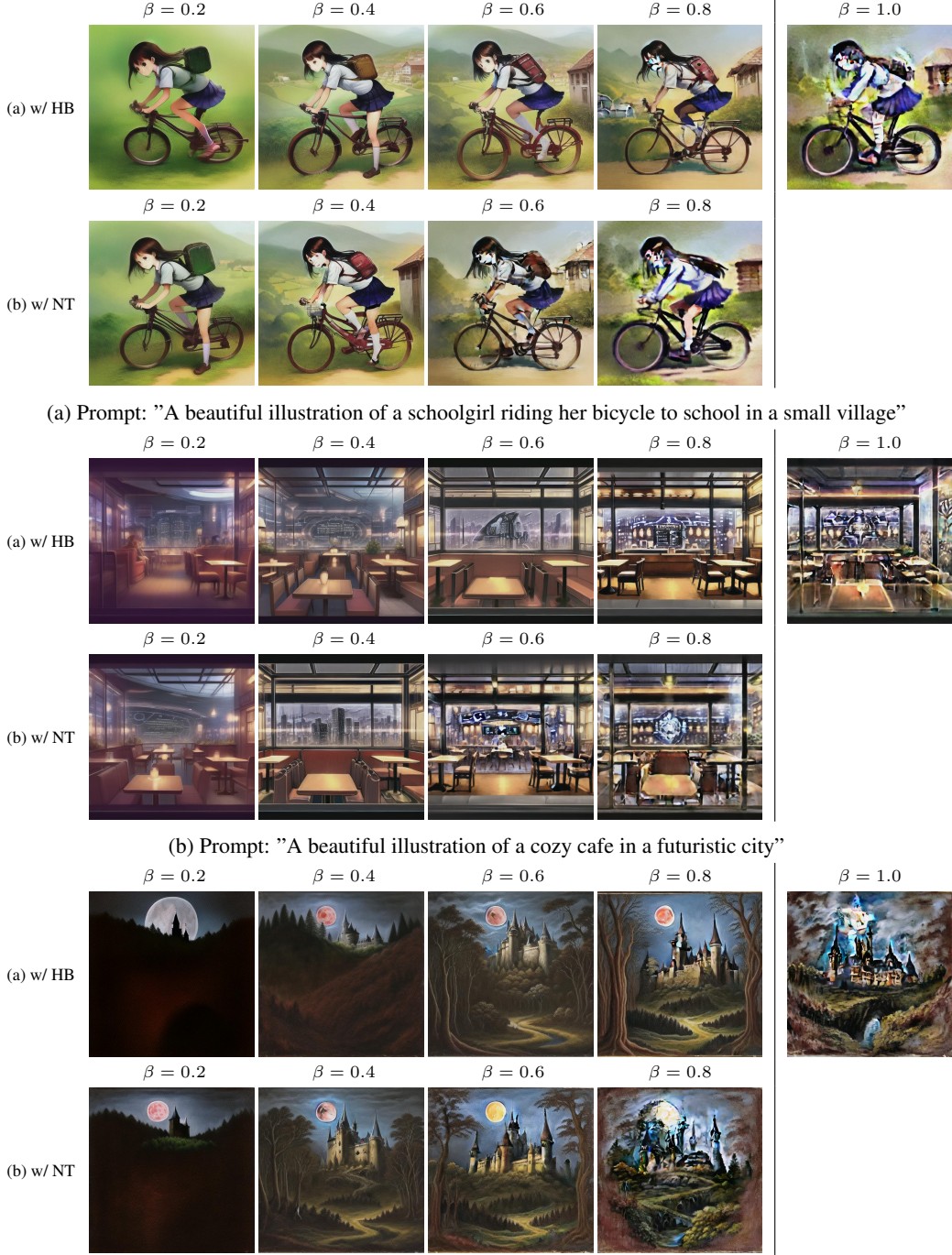

(a) Prompt: "A beautiful illustration of a schoolgirl riding her bicycle to school in a small village"

(b) Prompt: "A beautiful illustration of a cozy cafe in a futuristic city"

(c) Prompt: "A painting of an old European castle in a deep forest with a Blood Moon in the background"

Figure 37: Comparison between two variations of momentum: (a) Polyak's Heavy Ball (HB) and (b) Nesterov (NT). These momentum variations are applied to PLMS4 (Liu et al., 2022a) on a fine-tuned Stable Diffusion model called Anything V4 with 15 sampling steps and a guidance scale of 15. Both variations effectively reduce artifacts. However, the choice of the effectiveness parameter $\beta$ might differ due to the distinct shapes of their respective stability regions.

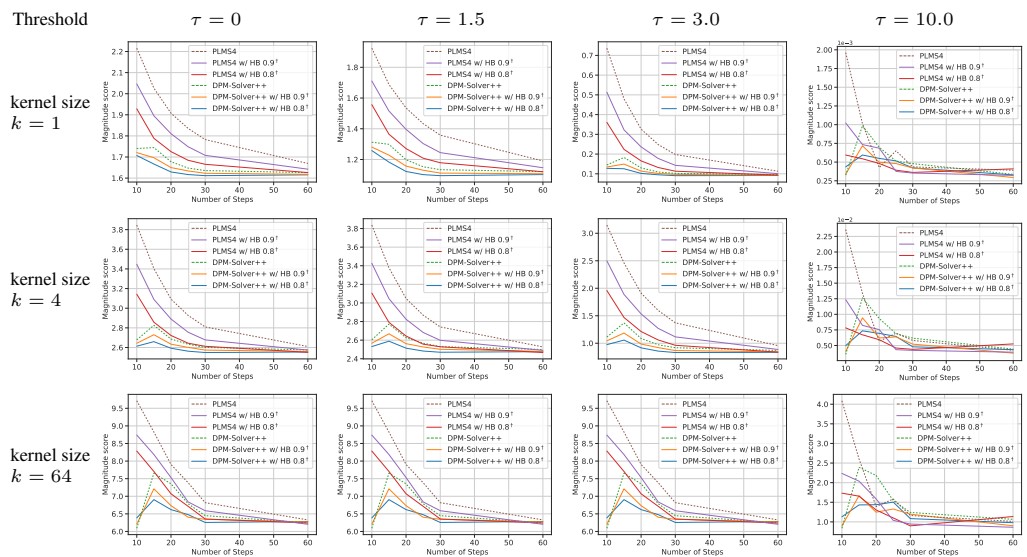

Figure 38: Comparison of magnitude scores on Anything V4 on different combinations of threshold $\tau$ and kernel size $k$ used in max-pooling. #samples = 160

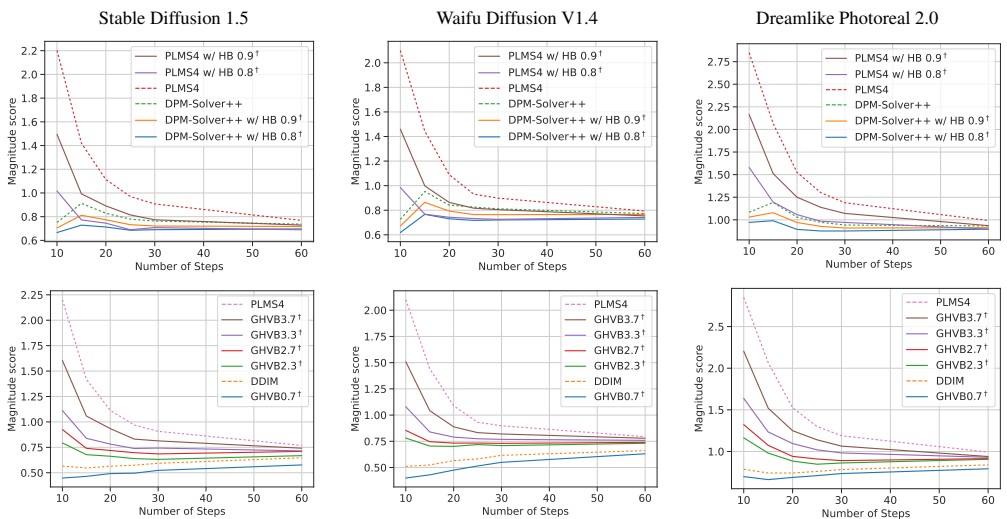

Figure 39: Comparison of magnitude scores in different diffusion models. #samples = 160

| Method | Number of steps | | | | | |
|---|---|---|---|---|---|---|
| | 10 | 15 | 20 | 25 | 30 | 60 |
| DPM | $1.113 \pm .090$ | $1.369 \pm .102$ | $1.087 \pm .083$ | $0.972 \pm .075$ | $0.919 \pm .068$ | $0.869 \pm .067$ |
| DPM w/ HB 0.8 | $0.974 \pm .078$ | $1.057 \pm .079$ | $0.916 \pm .072$ | $0.857 \pm .070$ | $0.831 \pm .067$ | $0.834 \pm .065$ |
| DPM w/ HB 0.9 | $1.043 \pm .082$ | $1.186 \pm .088$ | $0.986 \pm .075$ | $0.921 \pm .073$ | $0.867 \pm .068$ | $0.844 \pm .065$ |
| PLMS4 w/ HB 0.8 | $1.958 \pm .116$ | $1.469 \pm .105$ | $1.213 \pm .097$ | $1.060 \pm .087$ | $0.963 \pm .076$ | $0.838 \pm .063$ |
| PLMS4 w/ HB 0.9 | $2.499 \pm .118$ | $1.888 \pm .112$ | $1.534 \pm .112$ | $1.270 \pm .104$ | $1.116 \pm .091$ | $0.887 \pm .066$ |
| PLMS4 | $3.149 \pm .116$ | $2.460 \pm .116$ | $1.911 \pm .115$ | $1.597 \pm .116$ | $1.372 \pm .106$ | $0.957 \pm .075$ |

Table 8: 95% confidence intervals for the magnitude scores of HB (Figure 7)

| | | | Number of steps | | | |
|---|---|---|---|---|---|---|
| Method | 10 | 15 | 20 | 25 | 30 | 60 |
| DDIM | $0.844 \pm .076$ | $0.765 \pm .064$ | $0.728 \pm .060$ | $0.744 \pm .063$ | $0.761 \pm .062$ | $0.778 \pm .062$ |
| GHVB2.1 | $1.238 \pm .097$ | $0.924 \pm .072$ | $0.832 \pm .067$ | $0.820 \pm .066$ | $0.825 \pm .066$ | $0.829 \pm .064$ |
| GHVB2.3 | $1.291 \pm .102$ | $0.952 \pm .072$ | $0.872 \pm .070$ | $0.836 \pm .064$ | $0.831 \pm .066$ | $0.828 \pm .062$ |
| GHVB2.5 | $1.392 \pm .103$ | $1.016 \pm .081$ | $0.907 \pm .071$ | $0.851 \pm .069$ | $0.842 \pm .065$ | $0.845 \pm .063$ |
| GHVB2.7 | $1.514 \pm .105$ | $1.095 \pm .085$ | $0.968 \pm .077$ | $0.877 \pm .068$ | $0.864 \pm .067$ | $0.826 \pm .063$ |
| GHVB2.9 | $1.673 \pm .107$ | $1.203 \pm .090$ | $1.023 \pm .079$ | $0.934 \pm .075$ | $0.901 \pm .071$ | $0.835 \pm .063$ |
| PLMS4 | $3.149 \pm .116$ | $2.460 \pm .116$ | $1.911 \pm .115$ | $1.597 \pm .116$ | $1.372 \pm .106$ | $0.957 \pm .075$ |

Table 9: 95% confidence intervals for the magnitude scores of GHVB (Figure 8)

| | | | | Number of steps | | | |
|---|---|---|---|---|---|---|---|
| Method | 10 | 20 | 40 | 80 | 160 | 320 | 640 |
| DDIM | $0.584 \pm .034$ | $0.409 \pm .029$ | $0.304 \pm .029$ | $0.210 \pm .026$ | $0.139 \pm .022$ | $0.085 \pm .014$ | $0.048 \pm .009$ |
| GHVB1.1 | $0.592 \pm .034$ | $0.406 \pm .029$ | $0.295 \pm .030$ | $0.189 \pm .026$ | $0.113 \pm .020$ | $0.054 \pm .010$ | $0.019 \pm .005$ |
| GHVB1.3 | $0.609 \pm .035$ | $0.410 \pm .029$ | $0.276 \pm .029$ | $0.158 \pm .023$ | $0.086 \pm .017$ | $0.030 \pm .007$ | $0.009 \pm .003$ |
| GHVB1.5 | $0.624 \pm .036$ | $0.409 \pm .029$ | $0.261 \pm .029$ | $0.145 \pm .023$ | $0.067 \pm .014$ | $0.021 \pm .005$ | $0.006 \pm .002$ |
| GHVB1.7 | $0.645 \pm .037$ | $0.411 \pm .030$ | $0.254 \pm .028$ | $0.133 \pm .023$ | $0.053 \pm .011$ | $0.016 \pm .005$ | $0.004 \pm .002$ |
| GHVB1.9 | $0.663 \pm .037$ | $0.414 \pm .030$ | $0.246 \pm .028$ | $0.123 \pm .021$ | $0.044 \pm .009$ | $0.013 \pm .004$ | $0.003 \pm .001$ |
| PLMS2 | $0.676 \pm .038$ | $0.418 \pm .030$ | $0.246 \pm .028$ | $0.119 \pm .021$ | $0.041 \pm .009$ | $0.011 \pm .003$ | $0.003 \pm .001$ |

Table 10: 95% confidence intervals for L2 norm of GHVB (Figrue 11)

| | | | Number of steps ($k_{\text{new}}$) | | |
|---|---|---|---|---|---|
| Method | 40 | 80 | 160 | 320 | 640 |
| GHVB0.5 | $0.247 \pm .030$ | $0.235 \pm .029$ | $0.351 \pm .045$ | $0.450 \pm .057$ | $0.474 \pm .057$ |
| GHVB1.5 | $0.550 \pm .072$ | $0.717 \pm .086$ | $0.922 \pm .089$ | $1.337 \pm .102$ | $1.519 \pm .102$ |
| GHVB2.5 | $0.624 \pm .077$ | $1.121 \pm .115$ | $1.546 \pm .132$ | $1.906 \pm .153$ | $1.846 \pm .147$ |
| GHVB3.5 | $0.459 \pm .063$ | $0.920 \pm .107$ | $1.877 \pm .170$ | $1.960 \pm .147$ | $1.779 \pm .163$ |

Table 11: 95% confidence intervals for the numerical orders of convergence of GHVB (Figure 12)

| | Number of steps | | |
|---|---|---|---|
| Method | 15 | 30 | 60 |
| DPM-Solver++ | 2.49 | 4.84 | 9.54 |
| DPM-Solver++ w/ HB 0.9 | 2.49 | 4.84 | 9.54 |
| PLMS4 | 2.49 | 4.84 | 9.54 |
| PLMS4 w/ HB 0.9 | 2.46 | 4.79 | 9.43 |
| PLMS4 w/ NT 0.9 | 2.53 | 4.93 | 9.70 |
| GHVB3.9 | 2.50 | 4.84 | 9.54 |

Table 12: Comparison of the average sampling time per image (in seconds) when using different numbers of steps in Stable Diffusion 1.5 on a single NVIDIA GeForce RTX 3080. The time differences are marginal.

