# OpenReview forum: "Diffusion Sampling with Momentum for Mitigating Divergence Artifacts"
_ICLR.cc/2024/Conference — ICLR 2024 poster_

### Official Review · Reviewer_uXih · 2023-10-17

**Soundness:** 3 good
**Presentation:** 3 good
**Contribution:** 3 good
**Rating:** 8
**Confidence:** 4

**Summary:**

The paper introduces the Heavy Ball (HB) momentum into diffusion numerical methods to expand the stability regions. Meanwhile, the authors propose the high-order method, Generalized Heavy Ball (GHVB), to select the suitable method. Experiments show that the proposed HB and GHVB improves existing on both pixel-based and latent-based diffusion model in reducing artifacts and improving image quality.

**Strengths:**

1. The authors introduce the Heavy Ball (HB) momentum into existing diffusion methods to expand the stability regions. And they propose a high-order method, Generalized Heavy Ball (GHVB), to trade off between accuracy and artifact suppression.
2. The analyses are adequate. Through visualization and theoretical analysis, it is discovered that the small stability regions lead to model artifacts.
3. The experiments are extensive. The authors apply HB and GHVB on pixel-based and latent-based diffusion models to prove the effectiveness of the proposed method.
4. The authors also provide the code, which shows the solidness of the work.

**Weaknesses:**

The paper primarily experiments with 10 or more generation steps. But, it lacks analyses of extreme cases, such as one or two steps. It is suggested to evaluate the effectiveness of the proposed methods in these scenarios, e.g., one or two steps. For instance, the consistency model [1] performs well in one- and few-step generation. How effective is the method proposed in this paper compared with CM?

[1] Consistency Models, Song, Yang and Dhariwal, Prafulla and Chen, Mark and Sutskever, Ilya

**Questions:**

1. How effective are the proposed methods in extremely small generation steps, such as one or two?

---

> ### Author Response · Authors · 2023-11-16
> **Rebuttal for Reviewer uXih**
>
> > The paper primarily experiments with 10 or more generation steps. But, it lacks analyses of extreme cases, such as one or two steps. It is suggested to evaluate the effectiveness of the proposed methods in these scenarios, e.g., one or two steps. For instance, the consistency model [1] performs well in one- and few-step generation. How effective is the method proposed in this paper compared with CM?
> > How effective are the proposed methods in extremely small generation steps, such as one or two?
> - We appreciate the reviewer's suggestion to evaluate in extreme cases, particularly with one or two generation steps. Currently, every state-of-the-art high-order method relies on previous evaluations to better estimate the next evaluation point, and without a certain amount of previous evaluations, high-order methods will perform the same as low-order methods. In other words, all methods tend to yield similar results (see result in this [link](https://pic.in.th/image/TVMHCp)). Consequently, the performance of sampling becomes less dependent on the sampling method and more reliant on the model's effectiveness, especially when the number of steps is extremely low, such as one or two steps. So, when comparing with CM, any training-free methods will yield very similar, if not the same, result which can be seen in previous works [1,2].
> - It is true that methods like the consistency model can yield the extreme efficiency of one- or two-step generation. However, they require creating new models by either distilling old ones or training from scratch. This is contrary to our approach, which focuses on designing a training-free pipeline that can be applied to any existing models.
> - [2] Latent Consistency Models: Synthesizing High-Resolution Images with Few-Step Inference

---

> > ### Comment · Reviewer_uXih · 2023-11-22
> > **After rebuttal**
> >
> > Thank you for your rebuttal. The authors' analysis of extreme cases is reasonable, and therefore, a comparison with CM is not necessary. I maintain my original rating.

---

### Official Review · Reviewer_Z3BT · 2023-10-31

**Soundness:** 3 good
**Presentation:** 2 fair
**Contribution:** 3 good
**Rating:** 5
**Confidence:** 3

**Summary:**

This paper focuses on accelerating general diffusion sampling, where both unconditional and guided sampling are considered. Motivated by the observation that recent higher-order numerical methods would lead to diverging artifacts at lower sampling steps, the authors propose to incorporate heavy ball (HB) momentum into existing diffusion ODE solvers such as DPM++ and PLMS to mitigating their artifacts. In addition, an improved high-order version, namely generalized heavy Ball (GHVB) is also presented in this paper.  Experimental results have shown the effectiveness of this proposal.

**Strengths:**

1), Both pixel-based and latent diffusion models are considered in this paper.

2), The presentation is overall easy to follow.

3), Good practical extension to DPM++ and PLMS.

4), The literature over existing high order ODE solvers seems up to date.

**Weaknesses:**

1), The technical novelty behind this work seems to be not significant. The main techniques used in this paper are directly borrowed from Polyak’s heavy ball (HB) momentum method, a conventional optimization algorithm. Besides, the main improvements of this work are built based on DPM++ and PLMS.

2), While two methods are proposed in the same paper, it is unclear which one should be used under what circumstances. The paper only gives some vague statements without comprehensive comparison.

3), While guided diffusion sampling is considered, the effectiveness of the HB/GHVB under different scaling factor “s” is not well discussed.

**Questions:**

1), While the authors mentioned that the problem setup is more challenging in this paper than previous works, it is unclear what the challenges are. More discussions about why PLMS and DPM-Solver ++ perform worth than their original claims would strengthen this proposal.

2), Given that the 1000-Step DDIM’s sample is considered the benchmark, it would be reasonable to include evaluation metrics such as L2, LIPIS, and FID comparing HB/GHVB to DDIM, as depicted in Figure 11.

3), In Figure. 12, the authors attribute the inconsistency of GHVB 2.5 and 3.5 to estimated error or other sources of error without further justifications. It would be helpful to discuss this more for better understanding.

4), Seems the comparisons and discussions between HB and GHVB are not sufficient in the paper’s current state. There is no clear cut which method is better for both conditional and unconditional diffusion sampling.

---

> ### Author Response · Authors · 2023-11-20
> **Rebuttal for Reviewer Z3BT (part 1/2)**
>
> > The technical novelty behind this work seems to be not significant. The main techniques used in this paper are directly borrowed from Polyak’s heavy ball (HB) momentum method, a conventional optimization algorithm. Besides, the main improvements of this work are built based on DPM++ and PLMS.
>
> - Our GHVB, a high-order generalization of Polyak's heavy ball momentum method, is an original approach not borrowed from elsewhere. Though, we acknowledge that the HB (not GHVB) algorithm may be perceived as an application of existing concepts like momentum. Nonetheless, our distinct contribution is being the first to employ and analyze HB as an ODE solver within the context of diffusion sampling. Our study also provides novel findings resulting from extensive experiments and well-tested hypotheses, which serve as a basis for practical adoption
> - Additionally, our HB method may seem to build upon previous methods like DPM++ and PLMS because it is designed as an add-on specifically aimed at addressing their divergence artifact problem.
>
> > While two methods are proposed in the same paper, it is unclear which one should be used under what circumstances. The paper only gives some vague statements without comprehensive comparison.
> > Seems the comparisons and discussions between HB and GHVB are not sufficient in the paper’s current state. There is no clear cut which method is better for both conditional and unconditional diffusion sampling.
> - A direct comparison is presented in Figure 5, demonstrating that both HB and GHVB effectively reduce artifacts. However, HB exhibits a faster accuracy drop, leading to blurry images.  A discussion between these methods can be found in Section 7, emphasizing HB's role as an add-on for artifact reduction, while GHVB offers flexibility in method selection. Therefore, we advocate for choosing the high-order generalization method (GHVB) over first-order methods like HB. Ideally, the selection should prioritize the highest-order method whose stability region still covers all eigenvalues, as detailed in Appendix S, Topic Q2 (moved to Appendix T in the revision).
>
> > While guided diffusion sampling is considered, the effectiveness of the HB/GHVB under different scaling factor “s” is not well discussed.
> - Please see Figures 24-32 in Appendix N (moved to Fig. 25-33 in Appendix O), where we demonstrate the effectiveness of our HB and GHVB under various guidance scales for guided diffusion sampling.
>
> > While the authors mentioned that the problem setup is more challenging in this paper than previous works, it is unclear what the challenges are. More discussions about why PLMS and DPM-Solver ++ perform worth than their original claims would strengthen this proposal.
> - A short answer: The challenges here refer to the ability to speed up and reduce the number of sampling steps while avoiding divergence artifacts. All these algorithms generate artifacts, but only when the sampling steps drop below a certain threshold, which might not always be tested or demonstrated in their papers. In theory, eigenvalues and step size are primary factors leading to divergence in ODE solving. Reducing the step size is the most obvious way to make the problem more challenging, as highlighted in Section 1's second paragraph and depicted in Figure 1. Another approach involves altering sampling problems' eigenvalues, such as those using guidance sampling, higher guidance scales, LoRA fine-tuning, ControlNet, or the "Reference-Only" pipelines detailed in Appendices M and N (moved to N and O). These sampling problems tend to produce more artifacts at the same number of sampling steps compared to standard diffusion sampling, making them more challenging in our context.

---

> ### Author Response · Authors · 2023-11-20
> **Rebuttal for Reviewer Z3BT (part 2/2)**
>
> > Given that the 1000-Step DDIM’s sample is considered the benchmark, it would be reasonable to include evaluation metrics such as L2, LIPIS, and FID comparing HB/GHVB to DDIM, as depicted in Figure 11.
> - We agree with and appreciate your suggestion and will update the evaluation to use the 1000-step DDIM including Figure 11. According to the convergence theorem, numerical methods converge to the same exact solution with a high number of steps. Hence, this modification should not impact the final outcome.
>
> > In Figure. 12, the authors attribute the inconsistency of GHVB 2.5 and 3.5 to estimated error or other sources of error without further justifications. It would be helpful to discuss this more for better understanding.
> - The discrepancies observed in the plots depicted in Figure 12, which appear less consistent with the theoretical predictions, may come from various sources and reasons. One possible explanation is the limited precision of neural networks and the already very low errors for 640-step sampling of Figure 11 ($\sim 10^{-3}$). This scenario amplifies the sensitivity of the $e_{new} / e_{old}$ ratio to computational inaccuracies, even as small as $10^{-4}$.To illustrate this potential behavior, we demonstrate that errors from both the 999-step and 1,000-step DDIM can vary significantly by up to 0.0026. Theoretically, this difference should align much closer to zero. Additional sources of error include the time discretization nature of the diffusion network, which might not consistently across all number of steps (e.g., 320 vs 640). These errors are negligible when the total error is  large but can significantly compound and affect our *empirical approximation* of convergence order when dealing with very small errors.
> - [link to the experiment](https://colab.research.google.com/drive/11d5eXZHaG2qCJY_TgB8-zq1fagSmA7b8?usp=sharing)

---

### Official Review · Reviewer_DpnC · 2023-11-01

**Soundness:** 3 good
**Presentation:** 2 fair
**Contribution:** 3 good
**Rating:** 6
**Confidence:** 2

**Summary:**

This paper considers the artifacts problem of ODE/SDE-based diffusion sampling. Authors thought that the divergence artifacts are caused by the stability regions of high-order numerical methods for solving ODEs and proposed two solutions for expanding the stability regions of current diffusion numerical methods, called Heavy Ball (HB) momentum and Generalized Heavy Ball. And in the case of low-step sampling, the proposed methods are effective in reducing artifacts. But the actual improvement on diffusion sampling acceleration is unlear.

-------------
Post-rebuttal: I read the rebuttal and thanks for the authors' efforts. I would like to keep my score.

**Strengths:**

1.	The divergence artifacts problem is theoretically linked with the stability region of high-order numerical solvers for ODEs. The insight is very helpful for the design of diffusion sampling methods.
2.	To enlarge the stability region, authors proposed Heavy Ball (HB) and generalized Heavy Ball (GHVB) as two solution without any training. Experiments show that the divergence artifacts are great mitigated in a low-step sampling case.
3.	This paper is well organized and solid in theory.

**Weaknesses:**

1.	The proposed method should be compared with the state-of-art methods in reducing divergence artifacts if it is a big challenge in diffusion models.
2.	The stated motivation is diffusion model acceleration. Experiments are limited in comparing the results of few-step sampling, lacking clear numerical experiments in model acceleration.
3.	It seems that the proposed methods show superior performance only in extremely low sampling steps. In the case of decent image quality, the improvement on sampling step is unclear.

**Questions:**

1.	The main difference between HB and GHVB is that HB calculates the moving average after summing high-order coefficients, whereas GHVB calculates it before the summation. Why does such a difference lead a larger stability region?
2.	Can the divergence artifacts be solved or mitigated by improving the dynamic range of pixel?
3.	With additional training, what is the proposed methods’ complexity or cost?

---

> ### Author Response · Authors · 2023-11-16
> **Rebuttal for Reviewer DpnC (part 1/2)**
>
> > The proposed method should be compared with the state-of-art methods in reducing divergence artifacts if it is a big challenge in diffusion models.
> - As far as we understand, no other researchers have identified divergence artifacts as a problem in diffusion sampling. Therefore, we solely compare our approach with the current state-of-the-art diffusion sampling techniques outlined in both our main paper and appendix.
>
>
> > But the actual improvement on diffusion sampling acceleration is unclear
> > It seems that the proposed methods show superior performance only in extremely low sampling steps. In the case of decent image quality, the improvement on sampling step is unclear.
> > The stated motivation is diffusion model acceleration. Experiments are limited in comparing the results of few-step sampling, lacking clear numerical experiments in model acceleration.
> - We would like to first clarify the term 'acceleration’ in the context of our work. Here, acceleration refers to our ability to decrease the number of sampling steps while still producing good results without degeneration or divergence artifacts. For example, a sampling technique that produces artifacts using 10 sampling steps is considered slower than another that still remains artifact-free at step 10, when applied to the same diffusion model. In this context, the number of steps is a reasonable representation of speed, as the extra computation time for each step—due to computations related to momentum or other high-order aggregation—is negligible compared to the network evaluation time. A method that accelerates sampling by 25%, for example, means that the number of sampling steps can be reduced by 25% without the onset of artifacts."
> - Regarding your concern, please see Appendix M (moved to N in the revision) where we have provided comprehensive analyses of speed improvement, some of which is reproduced here. In Appendix M.1 (N.1), our experiments demonstrate that HB and GHVB reduce sampling time by 16.53% and 25.38%, respectively, compared to PLMS4 without momentum. Moreover, DPM-Solver++ with HB momentum achieves an 18.22% reduction in sampling time compared to the same algorithm without momentum. In Appendix M.2 (N.2), our experiments indicate significant reductions in sampling time from PLMS4, amounting to 32.40% and 42.92% with HB and GHVB, respectively. We believe these quantitative results show that our algorithms yield clear sampling speedups.
> - Additionally, our experiments encompass a wide range of numbers of sampling steps, not just extremely low ones. For instance, in Section 5.1, we experimented with step numbers ranging from 10 to 640. In Appendix M (moved to N), they range from 10 to 300 steps. In certain experiments requiring FID score computation, we limited the range due to computational resources. For examples, we use (1) the 10 to 30 range in Section 5.2 and Appendix I (moved to J) and (2) the 6 to 25 step range in Section 5.3 and Appendix J (moved to K). We don't consider these ranges to be extremely low.
>
> > The main difference between HB and GHVB is that HB calculates the moving average after summing high-order coefficients, whereas GHVB calculates it before the summation. Why does such a difference lead a larger stability region?
> - In the concluding paragraph of Section 2.2, we highlighted that higher-order methods tend to have smaller stability regions. Given HB's first-order convergence compared to GHVB's designed high-order convergence, HB naturally exhibits a larger stability region. To understand how formulation differences impact convergence orders, please refer to Section 4.2 and Appendix F (moved to G). Higher convergence orders indicate that more terms are matched in the Taylor expansion, resulting in a more accurate approximation. Notably, GHVB matches more terms than HB

---

> ### Author Response · Authors · 2023-11-16
> **Rebuttal for Reviewer DpnC (part 2/2)**
>
> > Can the divergence artifacts be solved or mitigated by improving the dynamic range of pixel?
>
> - The short answer is unlikely. We haven't found any evidence suggesting that improving the range of pixels can mitigate the divergence artifacts. Techniques like dynamic thresholding [1], which clamp the $x_0$ prediction within some thresholds, do not effectively prevent these artifacts (see below). Moreover, in Appendix I (moved to J), we've utilized a static thresholding technique [1], capping the $x_0$ value to the maximum and minimum pixel values at each iteration, but this method also doesn't alleviate the issue. As depicted in Figure 16, the artifacts persist even when their values are capped.
>
> - Here, we conducted a test involving dynamic thresholding specifically on Stable Diffusion. We utilized huggingface’s diffusers implemented on DPM-Solver++ from the provided [link](https://github.com/huggingface/diffusers/blob/v0.23.0/src/diffusers/schedulers/scheduling_dpmsolver_multistep.py#L74). The resulting image, displayed in ([Here](https://pic.in.th/image/T0jFOy)), did not demonstrate a reduction in divergence artifacts.
>
> [1] : Photorealistic Text-to-Image Diffusion Models with Deep Language Understanding (2022)
>
> > With additional training, what is the proposed methods’ complexity or cost?
> - Our method is training-free and applicable to any diffusion sampling. We believe there might be some typos in the question. In case the reviewer meant the complexity and/or cost induced by our sampling algorithms, the inclusion of one line of code for computing the moving average has a very small impact on sampling time, as detailed in Table 12 of Appendix R (moved to S). Both of our methods show no significant differences in sampling times compared to other methods. We hope this clarifies these aspects.

---

### Official Review · Reviewer_jPhx · 2023-11-01

**Soundness:** 3 good
**Presentation:** 3 good
**Contribution:** 3 good
**Rating:** 8
**Confidence:** 2

**Summary:**

This submission suggests to use higher order numerical scheme (heavy ball momentum coupled with higher order multi-step methods in numerical ODE) to compute the diffusion process in computer vision.

**Strengths:**

Authors' effort in experiments seem to be solid and thorough.
Authors have also been patient to review basics of stability concept in numerical ODEs.

**Weaknesses:**

I recommend that authors add a paragraph explaining what "sampling" means in the context of diffusion in the appendix, so that the content can be more self-contained. From what I understand about the main text, authors mean generating/inferring an image with trained diffusion models. This is not equivalent to the meaning of illustrating the distribution of all potentially generated images given underlying diffusion models.


I also suggest that authors make a table to list all used numerical formats, explicitly, either in the main text or in appendix, to generate images. In this way, readers can associate the listed methods in each table/figure with specific algorithms.
The current presentation stops at a conceptual derivation of discrete update format instead of concrete update formula. In a similar spirit, it will be also helpful for authors to detail the setup of the training paradigm (specifically, what the loss function is for training).

**Questions:**

- Are metrics "high-frequency error norm (HFEN)" [MR image reconstruction from highly undersampled k-space data by dictionary learning, Ravishankar and Bresler, 2011] and Structural Similarity Index (SSIM) potentially relevant to measure the divergence artifacts (section 5.1)? If yes, then reporting evaluation results in these two metrics can be helpful.

- Conceptually, I would like to understand better what authors mean by "classifier-guided diffusion sampling". What is the difference (conceptually and when it comes to implementation) between classifier-guidance and text-prompt based generation?

---

> ### Author Response · Authors · 2023-11-20
> **Rebuttal for Reviewer jPhx**
>
> > add a paragraph explaining what "sampling" means
> > add table list all used numerical formats, explicitly,
> > Detail about setup of the training paradigm
> - Thank you for the valuable feedback. We appreciate the suggestions. We are happy to add a section in the appendix to explain more about diffusion sampling, create a table summarizing numerical methods for diffusion solver, and include a section with detailed information on diffusion training. (added to Appendix A in the revision)
>
> > What is classifier-guided diffusion sampling? What is the difference (conceptually and when it comes to implementation) between classifier-guidance and text-prompt based generation?
> - We acknowledge the confusion and will address it by adding a section in the appendix to provide a clearer explanation about guided diffusion sampling.
> In short, classifier-guided diffusion sampling [1] utilizes another model, typically a classifier model, to guide the sampling process towards a desired direction. This is in contrast to classifier-free sampling [2], where the diffusion model itself directs the outcome. Text-to-Image generation can be achieved through both classifier-guided sampling, as seen in CLIP-guided sampling, or classifier-free sampling. However, in approaches like Stable Diffusion, the sampling is predominantly considered classifier-free.
>
> - [1] : Diffusion Models Beat GANs on Image Classification
> - [2] : Classifier-Free Diffusion Guidance
>
> > Are metrics HFEN, SSIM potentially relevant to measure the divergence artifacts?
> - In contrast to the magnitude score, the HFEN and SSIM metrics do not directly quantify the extent of divergence artifacts in the results. However, they can detect divergence by comparing the differences between individual samples and their convergence results, akin to the use of l2 and LPIPS metrics, which we have employed throughout our paper. We appreciate this suggestion and will include these measurements in our paper.

---

### Meta-Review · Area_Chair_pBEi · 2023-12-14

**Metareview:**

The paper introduces a family of higher order diffusion model solvers, based on the heavy ball method. The work is motivated by the desire to generate high-quality samples with limited computation. One promising family of approaches treats the diffusion model as an ODE and applies higher order solvers, with an aim toward achieving faster convergence. The paper observes that these approaches break down when one attempts to radically reduce the number of steps and correspondingly increase the step size. The proposed explanation is that these methods are being operated outside of their ``stability region’’, i.e., the step size is too large relative to the eigenvalues of the Jacobian at the target solution. This assertion is justified indirectly, by observing that in latent diffusion models, sampling artifacts at large step size co-occur with blow-up in the values of the latent variables. The paper proposes a heavy ball solver, and demonstrates numerically that different choices of the the momentum parameter can reshape the stability region, making it encompasss more of the real axis. The paper also introduces generalized heavy ball methods, which integrates momentum into higher order solvers.

The reviewers produced a mixed, but mostly positive evaluation of the paper (8865). On the positive side,
  + reviewers found the paper’s link between sampling artifacts and divergence insightful, and praised the paper's clarity
  + found the proposed  heavy ball method to be a simple, practical enhancement to existing deterministic diffusion models (DPM++ and PLMS).
  + Reviewers also found the paper to be clearly written, with a clear exposition of the concept of stability radius.
  + Experiments show that the proposed enhancements do improve DPM++ and PLMS, and do suppress artifacts

Concerns included:
- the paper proposes a family of methods without clear guidance on which to choose
- The empirical link between sampling artifacts and divergence is not 100% clear, since divergence is measured indirectly through the size of the latent variables
- The experiments in the initial submission are limited to small numbers of steps (where the discrepancies between methods may be maximized).
- Questions around technical novelty of the HB / GHB method, which could be addressed by better contextualizing the paper's proposals within the broader literature on ODE solvers

The authors response addressed some of these issues (esp, experimental limitations); the proposed momentum methods serve as a simple practical enhancement for existing deterministic diffusion models.

**Justification For Why Not Higher Score:**

While the paper provides a simple approach to improving the performance of deterministic diffusion models, it would be stronger with a clearer contextualization of the methods within both the diffusion model and broader ODE solver literatures (cf, reviewer comments on novelty), and clearer practical guidance on when to prefer GHB vs HB.

**Justification For Why Not Lower Score:**

The proposed momentum methods serve as a simple, practical enhancement for existing deterministic diffusion models, improving their accuracy / computation tradeoffs. The paper is clearly written and well motivated.

---

### Decision · Program_Chairs · 2024-01-16

Accept (poster)